# Distinct oligodendrocyte populations have spatial preference and different responses to spinal cord injury

Elisa M. Floriddia [1✉], Tânia Lourenço[1,2,3], Shupei Zhang[1], David van Bruggen [1], Markus M. Hilscher [4,5], Petra Kukanja [1], João P. Gonçalves dos Santos[1], Müge Altınkök[1], Chika Yokota[4], Enric Llorens-Bobadilla [6], Sara B. Mulinyawe[7], Mário Grãos [2,3], Lu O. Sun [7,8], Jonas Frisén[6], Mats Nilsson [4] & Gonçalo Castelo-Branco [1,9✉]

Mature oligodendrocytes (MOLs) show transcriptional heterogeneity, the functional consequences of which are unclear. MOL heterogeneity might correlate with the local environment or their interactions with different neuron types. Here, we show that distinct MOL populations have spatial preference in the mammalian central nervous system (CNS). We found that MOL type 2 (MOL2) is enriched in the spinal cord when compared to the brain, while MOL types 5 and 6 (MOL5/6) increase their contribution to the OL lineage with age in all analyzed regions. MOL2 and MOL5/6 also have distinct spatial preference in the spinal cord regions where motor and sensory tracts run. OL progenitor cells (OPCs) are not specified into distinct MOL populations during development, excluding a major contribution of OPC intrinsic mechanisms determining MOL heterogeneity. In disease, MOL2 and MOL5/6 present different susceptibility during the chronic phase following traumatic spinal cord injury. Our results demonstrate that the distinct MOL populations have different spatial preference and different responses to disease.

[1] Laboratory of Molecular Neurobiology, Department Medical Biochemistry and Biophysics, Karolinska Institutet, Biomedicum, 17177 Stockholm, Sweden. [2] Biocant, Technology Transfer Association, Cantanhede, Portugal. [3] Centre for Neuroscience and Cell Biology (CNC), University of Coimbra, Coimbra, Portugal. [4] Science for Life Laboratory, Department of Biophysics and Biochemistry, Stockholm University, 17165 Solna, Sweden. [5] Cartana AB, Nobels väg 16, 17165 Solna, Sweden. [6] Department of Cell and Molecular Biology, Karolinska Institutet, Biomedicum, 17177 Stockholm, Sweden. [7] Department of Neurobiology, Stanford University School of Medicine, Stanford, CA 94305, USA. [8] Department of Molecular Biology, University of Texas Southwestern Medical Center, Dallas, TX 75390, USA. [9] Ming Wai Lau Centre for Reparative Medicine, Stockholm Node, Karolinska Institutet, 171 77 Stockholm, Sweden. ✉email: elisa.floriddia@ki.se; goncalo.castelo-branco@ki.se

Myelin is produced by oligodendrocytes (OLs); it is highly enriched in lipids (80% of its dry mass), insulates (allowing electrical conduction) and metabolically supports axons[1]. Myelin synchronizes the impulse traffic between distant regions. This is a critical role to guarantee optimal motor, sensory, and higher-order cognitive functions[2]. For instance, cortical electrical impulses can be synchronized. Indeed, the conduction time between the left and right brain hemispheres is 30 ms, while it is 150–300 ms through unmyelinated short projecting fibers in the same hemisphere, so that impulses from different neurons may reach their target neurons in a coordinated manner when needed[3]. In addition, myelin proteins directly control synapse formation by inhibiting axonal sprouting[3]. Myelination is important during development and remains plastic throughout life. In fact, myelin is continuously produced and modulated by neuronal activity, allowing learning of new skills[4].

We have recently reported that the OL lineage is heterogeneous, as it is composed of transcriptionally distinct subpopulations/states during development and disease[5–7]. The heterogeneity of the OL lineage does not lie exclusively within the OL transcriptome. Indeed, developmentally distinct OPC pools generate OL lineage cells with different abilities to respond to demyelination[8]. Mature OLs (MOLs) form myelin internodes of various length and thickness, even along the same axon[9,10], and show properties regulated at both cell-autonomous and non-autonomous levels[4,11].

In this study, we asked whether the spatial distribution of MOLs might correlate with distinct OL functional states or subtypes. Single-cell RNA-sequencing (scRNAseq) unveiled the transcriptional heterogeneity of the OL lineage and suggested differential enrichment of MOLs in different regions of the central nervous system (CNS)[5]. The prediction of region enrichment based on scRNAseq can be biased by technical limitations, such as differences in viability of cell subpopulations during tissue dissociation[12].

In this study, we used RNAscope in situ hybridization (ISH) and in situ sequencing (ISS)[13] to determine the distribution of OL lineage populations. We found that distinct MOL populations present spatial preference in the CNS. The origin of distinct MOL populations is independent of intrinsic developmental mechanisms, but plausibly driven by extrinsic signals, such as neuronal electrical activity or other cues in the environment where MOLs reside. Furthermore, distinct MOLs present different responses to traumatic spinal cord injury (SCI) in the chronic phase, but not in the acute phase following injury or during demyelination in experimental autoimmune encephalomyelitis (EAE), a mouse model of multiple sclerosis. Our study paves the way for a more sophisticated understanding of the MOL populations-specific functional roles in development, health, and disease, possibly allowing better targeting of the OL subtypes to achieve regeneration and repair of the central nervous system.

## Results

### MOL2 and MOL5/6 show different spatial preference in the mouse brain and spinal cord.
We assessed the distribution of the OL lineage within white matter (WM) and gray matter (GM) of the brain and dorsal spinal cord in situ (Supplementary Fig. 1a, b). We performed immunohistochemistry (IHC) and RNAscope ISH for *Sox10* as a pan marker of the OL lineage and analyzed confocal images of the corpus callosum (WM), somatosensory cortex (GM) and the dorsal spinal cord (GM and WM) (Supplementary Fig. 1a, b). We analyzed the images with a custom automated pipeline (CellProfiler; Supplementary Fig. 1c–e).

*Ptprz1* (receptor-type tyrosine-protein phosphatase zeta 1) is a marker of OPCs and committed OPCs (COPs)[5,6]. *Sox10*+-*Ptprz1*+ cells presented a homogeneous distribution across the analyzed regions (Fig. 1a–c, m, Supplementary Fig. 1f). As expected, we observed a decreased contribution of OPCs/COPs to the OL lineage from juvenile to adulthood, especially in the somatosensory cortex (Fig. 1c, m, Supplementary Fig. 1f). Among the six transcriptionally distinct mature oligodendrocyte populations previously described, MOL1, MOL2, and MOL5/6 present the most distinct gene marker modules[5–7]; therefore, we analyzed their spatial distribution in the mouse central nervous system. *Egr2* (Early Growth Response 2; also known as *Krox20*) is expressed specifically by MOL1 in the OL lineage[5]. We observed a statistically non-significant difference of MOL1 distribution within the analyzed regions and across ages (Fig. 1d–f, m and Supplementary Fig. 1g). *Klk6* (Kallikrein Related Peptidase 6) is a distinct marker specific for MOL2 (Supplementary Fig. 1j)[5–7,14]. *Klk6* has been previously associated with demyelinating pathology in EAE and SCI[15,16]. Strikingly, we observed that *Klk6*+ MOL2 is a population specifically enriched in the dorsal spinal cord and almost absent in cortex and corpus callosum (Fig. 1g–i, m and Supplementary Fig. 1h). In contrast, MOL5 and 6 populations (MOL5/6) that exhibit high expression levels of *Ptgds* (Prostaglandin D2 Synthase) (Supplementary Fig. 1j)[5–7] showed a dynamic contribution to the OL lineage, increasing along time and following the myelination temporal pattern. Indeed, at juvenile (P20), MOL5/6 are more abundant in the dorsal spinal cord, where myelination is visible early on after birth (P5-6), compared to cortex and corpus callosum, where myelination is visible around P10-15. In adulthood (P60), MOL5/6 is the main population contributing to the OL lineage in both brain and spinal cord, being most abundant in the corpus callosum (Fig. 1j–m and Supplementary Fig. 1i). We confirmed the described spatial preference based on one population-specific marker detected with RNAscope ISH using ISS. ISS is a sequencing technology that allows to enquire the simultaneous expression of multiple RNAs in tissue sections[13]. We took advantage of the higher multiplexing power of ISS compared to RNAscope ISH to detect OPC/COP, MOL1, MOL2, and MOL5/6 based on the combination of multiple marker genes (Supplementary Fig. 2, 3 and Supplementary Data 1). MOL2 are enriched in *Anxa5, Hopx*, and *Klk6* and MOL5/6 have increased expression of *Ptdgs*, and also express *Grm3* and *Car2* (Supplementary Fig. 1j)[5]. ISS indicates the number of *Sox10*+/*Plp1*+ cells also positive for *Anxa5*/ *Klk6* or *Anxa5*/*Hopx* (MOL2) in the brain is lower than in the spinal cord, unlike *Sox10*+/*Plp1*+ cells positive for *Ptgds*/*Car2* and *Ptgds*/*Grm3* (MOL5/6) (Supplementary Fig. 3a and Supplementary Data 1 and 2). Furthermore, we observed that MOL5/6 are also increased with age in the cortex and corpus callosum, and spinal cord (Supplementary Fig. 3a, b and Supplementary Data 1 and 2). Altogether, the ISS data confirmed the spatial preference and distribution of MOL2 and MOL5/6 we observed by RNAscope ISH. In summary, here we described the spatial preference of the most abundant OL lineage populations and show these populations can be correctly identified by individual as well as multiple specific marker genes we carefully selected based on our previous scRNAseq characterization of the lineage[5–7,17].

### MOL2 preferentially reside in the WM and sensory tracts of the spinal cord, unlike MOL5/6.
Given the enrichment of MOL2 in the dorsal spinal cord compared to the analyzed brain regions (Fig. 1i, m), we further investigated whether MOL2 and MOL5/6 spatial preference relates to distinct regions and functional tracts in the spinal cord (Fig. 2a). Strikingly, we observed an

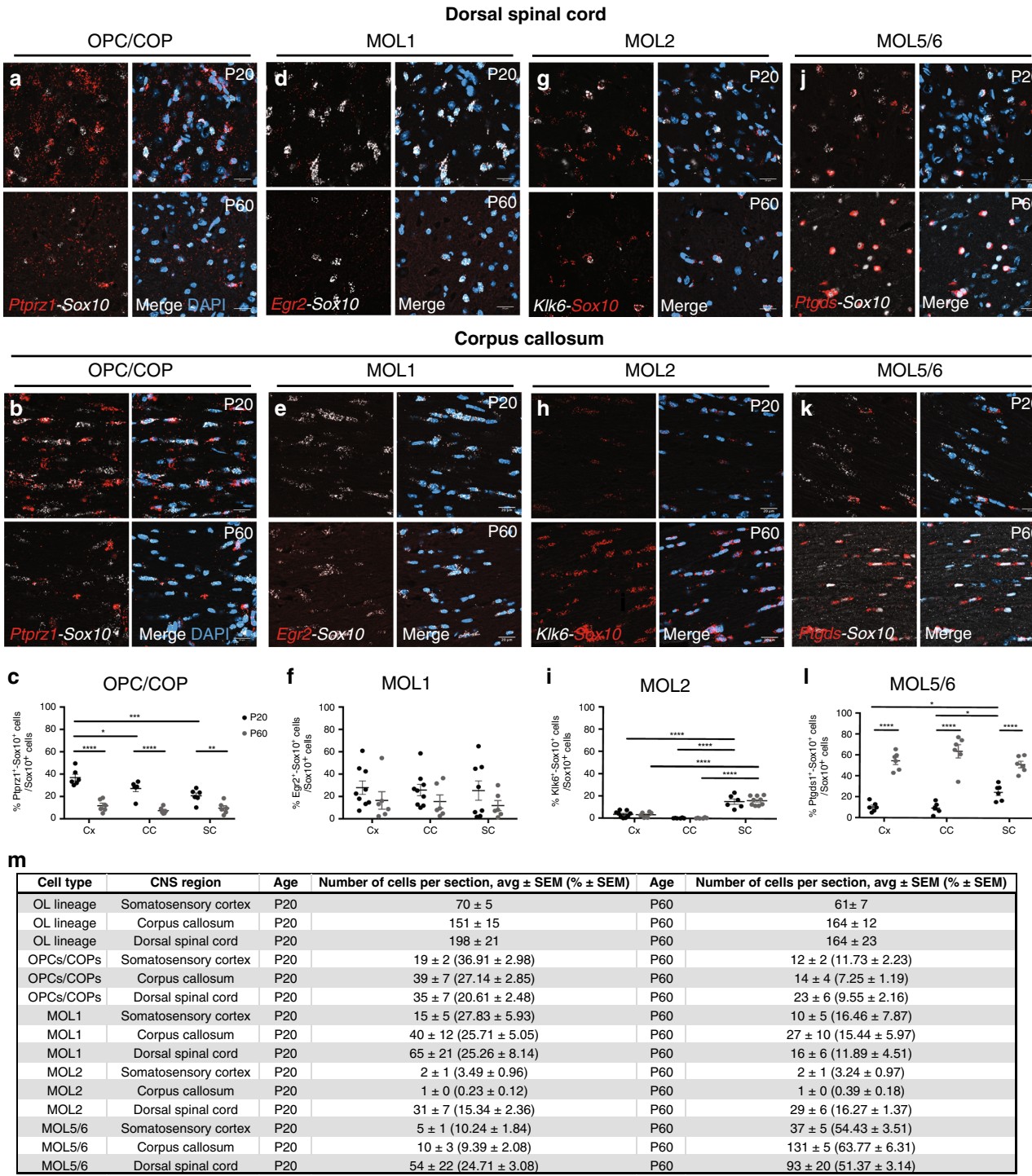

**Fig. 1 Specific mature OL populations have spatial preference in the juvenile and adult central nervous system. a–k** Confocal representative images of the distribution of OPC/COP (*Ptprz1*[+] OL lineage cells, **a** and **b**), MOL1 (*Egr2*[+] OL lineage cells, **d** and **e**), MOL2 (*Klk6*[+] OL lineage cells, **g** and **h**), and MOL5/6 (*Ptgds*[+] OL lineage cells, **j** and **k**) in the juvenile (P20) and adult (P60) dorsal spinal cord (dorsal horn gray matter, **a, d, g, j**) and corpus callosum (**b, e, h, k**). Scale bar = 20 μm. **c, f, i, l** Quantification of the OPC/COP (**c**), MOL1 (**f**), MOL2 (**i**), MOL5/6 (**l**) distribution in the cortex, corpus callosum, and dorsal spinal cord (dorsal horn and dorsal white matter) in juvenile (P20) and adulthood (P60). Percentage of the population is calculated on the total number of OL lineage cells (Sox10[+] cells) in the analyzed region. Data are presented as mean ± SEM. *n* = 6 animals per condition, and can be assessed in the Source Data file. Black circles—P20; Gray circles—P60; Asterisks indicate a significant difference between conditions (**p* ≤ 0.05, ***p* ≤ 0.01, ****p* ≤ 0.001, *****p* ≤ 0.0001, two-way ANOVA with Sidak's correction). Exact *p* values are reported in the Source Data file. **m** Quantification of the OL subpopulations contribution to the OL lineage. Average number and percentage of the OL lineage cells (Sox10[+]), OPC-COPs (*Ptprz1*[+] OL lineage cells), MOL1 (*Egr2*[+] OL lineage cells), MOL2 (*Klk6*[+] OL lineage cells), and MOL5/6 (*Ptgds*[+] OL lineage cells) contribution to the OL lineage (Sox10[+] cells) in the regions of interest in juvenile (postnatal day (P) 20) and adulthood (P60). We imaged 0.3, 0.13, and 0.4 mm[2] of the sensorimotor cortex, corpus callosum, and dorsal spinal cord per tissue section, respectively. Minimum three sections per animal were analyzed. Data are presented as mean ± SEM. *n* = 4−9 animals per condition, and can be assessed in the Supplementary Source Data file. Blue staining in merged images correspond to 4',6-diamidino-2-phenylindole (DAPI) staining. Cx cortex, CC corpus callosum, SC spinal cord. OPC oligodendrocyte progenitor cell, COP committed OPC, MOL mature oligodendrocyte. Source data are provided as a Source Data file.

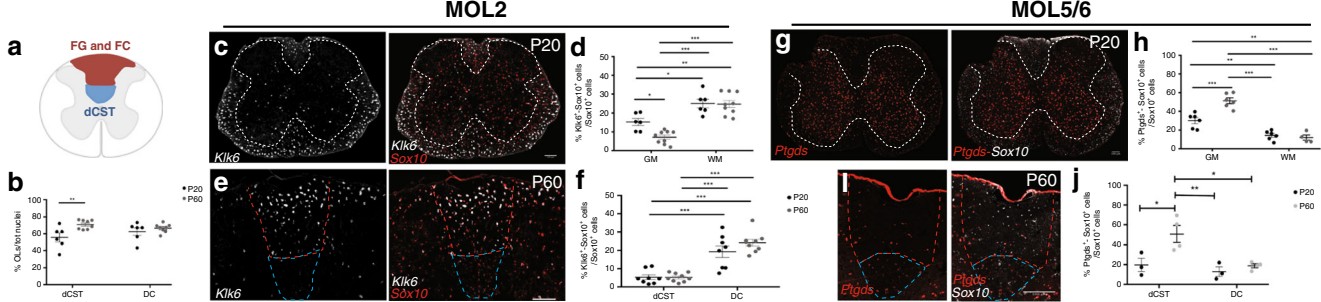

**Fig. 2 MOL2 and MOL5/6 are specifically enriched in adjacent regions of the juvenile and adult spinal cord. a** Schematics of a coronal section of the spinal cord. Highlighted in blue and red the dorsal funiculi (white matter region) where ascending (FG and FC) and descending (dCST) tracts run, respectively. **b** Percentage of the OL lineage cells (Sox10+ cells) calculated on the total number of nuclei shows an enrichment of the OL lineage in the dCST with age. Data are presented as mean ± SEM. n = 6−8 animals per condition, and can be assessed in the Source Data file. **c, e** Confocal representative images show the enrichment of MOL2 (Klk6+ OL lineage cells) in the white matter of the spinal cord (**c**) at postnatal day (P) 20 and at the level of the dorsal columns at P60 (**e**). **g, i** Confocal representative images show the enrichment of MOL5/6 (Ptgds+ OL lineage cells) in the gray matter of the spinal cord at P20 (**g**) and in the dorsal corticospinal tract at P60 (**i**). Scale bar = 100 μm. **d, f, h, j** Quantification of the MOL2 and MOL5/6 distribution in the white and gray matter of the spinal cord (**d, h**) and dorsal funiculi (**f, j**) in juvenile (P20) and adulthood (P60). Percentage of the population is calculated on the total number of OL lineage cells (Sox10+ cells) in the analyzed region. Data are presented as mean ± SEM. n = 3−9 animals per condition, and can be assessed in the Source Data file. Asterisks in all the panels in this figure indicate a significant difference between conditions (*p ≤ 0.05, **p ≤ 0.01, ***p ≤ 0.001, two-way ANOVA with Sidak's correction with multiple comparisons). Exact p values are reported in the Source Data file. Black circles—P20; Gray circles—P60. GM gray matter, WM white matter, FG fasciculi gracilis, FC fasciculi cuneatus, dCST dorsal corticospinal tract. MOL mature oligodendrocyte. Source data are provided as a Source Data file.

enrichment of MOL2 in the WM compared to the GM of the spinal cord (Fig. 2c, d and Supplementary Data 3). On the contrary, the MOL5/6 population is enriched in the GM compared to the WM of the spinal cord (Fig. 2g, h and Supplementary Data 3). Additionally, we observed that the MOL2 population decreases in the GM of the spinal cord from juvenile to adulthood (Fig. 2d and Supplementary Data 3). Also in this instance, we confirmed the MOL spatial preference by ISS analysis. Indeed, Ptgds+/Sox10+/Plp1+ or Car2+/Sox10+/Plp1+ cells (MOL5/6) are present in higher proportions in GM compared to WM, while Anxa5+/Sox10+/Plp1+, Hopx+/Sox10+/Plp1+. or Klk6+/Sox10+/Plp1+ cells (MOL2) are more abundant in the WM compared to the GM of the spinal cord at P20 (Supplementary Fig. 3c and Supplementary Data 1). Furthermore, analysis of a recent spatial transcriptomics[18] study[19] (https://als-st.nygenome.org/) investigating gene expression changes following amyotrophic lateral sclerosis indicates a similar spatial distribution of Klk6 and Hopx (MOL2) as well as Ptgds (MOL5/6) and Il33 genes (MOL6) (Supplementary Fig. 3e) as we report here. Despite the lack of focus on the heterogeneity of the OL lineage, the spatial distribution of these genes, characteristic of MOL populations, shown by Maniatis et al.[19] and our spatial analysis focused on the OL lineage populations presented here give confidence that MOL2 and MOL5/6 populations have distinct distributions across the CNS in vivo.

In accordance with previous reports[20–24], in the dorsal white matter (dorsal funiculi), the OL lineage density around the ascending sensory tracts (formed of the fasciculi gracilis and cuneatus) and dorsal corticospinal tract (motor tracts) correlates with the development of myelination. Indeed, we observed an increase in the OL lineage in the dorsal corticospinal tract with age (Fig. 2b and Supplementary Data 3) and constant distribution in the sensory tracts (Fig. 2b and Supplementary Data 3). More importantly, MOL2 are preferentially found where the sensory tracts run (Fig. 2e, f), while MOL5/6 preferentially localize in the dorsal corticospinal tract (Fig. 2i, j and Supplementary Data 3). In the dorsal funiculi, similarly to our observations in the GM and WM of the spinal cord, MOL2 and MOL5/6 show adjacent enriched distribution and spatial preference.

**Ventrally- and dorsally derived OPCs can give rise to similar MOLs.** The OL lineage derives from distinct progenitor domains and developmental stages[25], which might have a role in the specification of OPCs into distinct MOL populations and therefore a role in the observed spatial preference. To investigate this possibility, we first isolated the OL lineage from Emx1::Cre-Sox10::GFP-TdTom mice[26] and performed scRNAseq on ventrally (eGFP+) and dorsally (TdTom+) derived OL lineage cells. We analyzed the P60 corpus callosum (Fig. 3a) since this region has representation of OLs derived from both the cortical plate (dorsal domain) and the lateral and medial ganglionic eminences (ventral domains of the embryonic forebrain)[26]. As expected[26], a greater proportion of TdTom+ OL lineage cells was obtained (Fig. 3a, c and Supplementary Data 4). Graph-based clustering (Seurat)[27,28] of 2853 OL lineage cells and label transfer from the Marques et al. scRNA-Seq dataset[5] led to the identification of the previously identified OL lineage subpopulations (Fig. 3b and Supplementary Fig. 4a, b)[5,17]. Importantly, the contribution of ventrally- and dorsally derived OL lineage cells to each cluster was comparable (Fig. 3c, d), suggesting that the developmental waves have similar potential to generate the transcriptionally distinct OL lineage subpopulations.

In the brain, the ventrally derived and dorsally derived OL lineage cells arise during embryonic and early-postnatal development, respectively[25,29]. We thus further performed lineage tracing using the Pdgfrα::CreERᵀᴹ-loxP-RCE (Z/EG) mouse model[30] to fate map the progeny of pre-OPCs appearing at E13.5[6] or OPCs at P3-5 (Fig. 3e). Postnatal recombination of the Pdgfrα::CreERᵀᴹ-GFP mice results in the assessment of OPCs arising at both postnatal stages and remaining OPCs, while E12.5 recombination allows the assessment of ventral pre-OPCs arising from the first developmental wave. In the spinal cord, ventrally derived and dorsally derived OL lineage cells arise from two distinct embryonic stages of development (E13.5 and E15.5)[31]; therefore, our recombination strategy at E12.5 will label the pre-OPCs appearing at E13.5 in the spinal cord, while recombination at P3-5 will label OPCs derived from both embryonic timepoints that have not undergone differentiation yet and OPCs of postnatal origin (P0) (Fig. 3e), respectively.

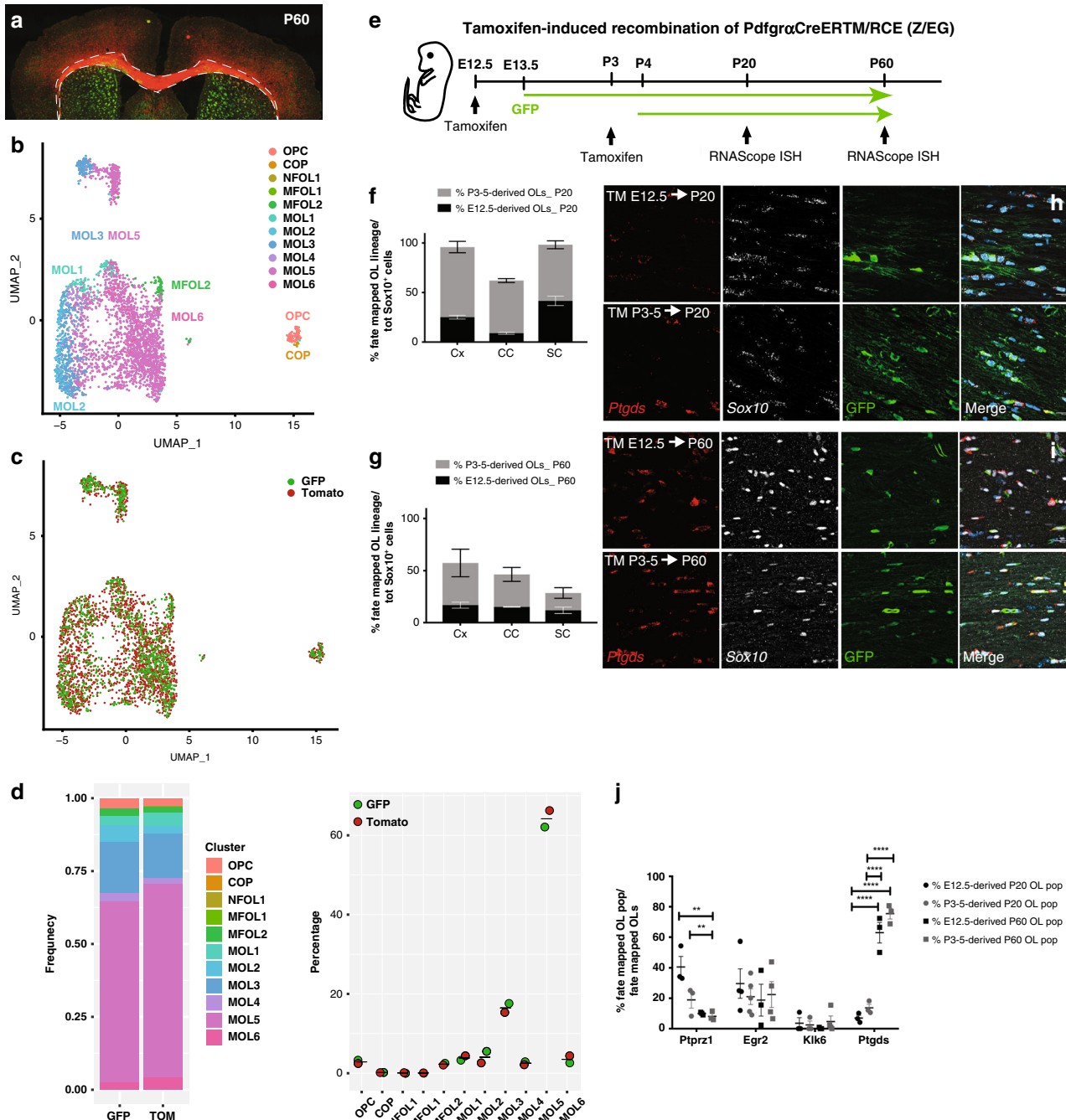

**Fig. 3 The developmental origin does not specify OPCs into distinct MOL populations. a** Confocal representative images of the postnatal day (P)60 brain from Emx1::Cre-SOX10::GFP-TdTom mouse. Dashed outline highlights the corpus callosum and dissected region used for scRNAseq. **b, c** Uniform Manifold Approximation and Projection (UMAP) plots showing the OL lineage composition determined by graph-based clustering (Seurat)[27,28] and integration cells with the Marques et al. scRNA-Seq dataset[5] (**b**) and the TdTom+ and GFP+ OL lineage cells contribution to the clusters (**c**). **d** Frequency distribution of the TdTom+ and GFP+ OL lineage cells forming the major clusters. $n = 2853$ cells. **e** Schematic overview of the fate mapping experimental design. Green arrows show the GFP expression timeline 24 h delayed from the time of tamoxifen injection. **f, g** Percentage of the fate mapped OL lineage cells (Sox10+-GFP+ cells) derived by OPCs labeled at E12.5 and P3-5 calculated on the total number of OL lineage cells (Sox10+ cells) at juvenile (**f**) and adulthood (**g**). Data are presented as mean ± SEM. $n = 4-5$ (**f**) and $3-5$ (**g**) animals per condition, and can be assessed in the Source Data file. **h, i** Confocal representative images show the MOL5/6 (Ptgds+-GFP+ OL lineage cells differentiated from pre- (TM E12.5) or postnatal (TM P3-5) OPCs in the P20 (**h**) and P60 (**i**) corpus callosum. Scale bar = 20 μm. **j** Percentages of the fate mapped OPCs-COPs (Ptprz1+-GFP+ OL lineage cells), MOL1 (Egr2+-GFP+ OL lineage cells), MOL2 (Klk6+-GFP+ OL lineage cells), and MOL5/6 (Ptgds+-GFP+ OL lineage cells) populations are calculated on the total number of fate mapped OL lineage cells (Sox10+-GFP+ cells) in the juvenile and adult corpus callosum. Data are presented as mean ± SEM. $n = 3-5$ animals per condition, and can be assessed in the Source Data file. Asterisks indicate a significant difference between conditions (**$p \leq 0.01$, ****$p \leq 0.0001$, two-way ANOVA with Sidak's correction). Exact $p$ values are reported in the Source Data file. TM tamoxifen, OPC oligodendrocyte progenitor cell, COP committed OPC, NFOL newly formed oligodendrocyte, MFOL myelin forming oligodendrocyte, MOL mature oligodendrocyte, TdTom tandem duplicated tomato, GFP green fluorescent protein. Source data are provided as a Source Data file.

Our recombination strategy labeled the majority of the lineage in the juvenile CNS (95.43 ± 6.22% of the Sox10[+] cells were positive for the eGFP reporter in the cortex, 62.24 ± 2.35% in corpus callosum, and 96.05 ± 1.15% spinal cord, Fig. 3f), similarly to previous reports[5,6,26,32]. We also observed sparser labeling in adulthood (62.38 ± 19.9% of the Sox10[+] cells were positive for the eGFP reporter in the cortex, 44.16 ± 9.04% in the corpus callosum, and 24.32 ± 8.26% in the dorsal horns; Fig. 3g). The considerable contribution and expansion of non-recombined late embryonic (E15.5 derived) and adult OPCs (GFP[−] generated within the process of life-long continuous addition of new myelinating OLs, for myelin homeostasis and in response to experience) might explain the relatively low labeling we observed in the juvenile corpus callosum and in all analyzed regions during adulthood[33,34]. Importantly, we observed that a subset of E12.5 pre-OPCs and their progeny do not disappear later in life in any of the analyzed regions (Fig. 3f, g and Supplementary Fig. 5), unlike previously reported[25,26,29].

Consistent with our scRNA-Seq data, we did not observe significant difference in the relative contribution of the two developmental waves to the generation of MOL1, MOL2, and MOL5/6 populations in the brain and spinal cord (Fig. 3h, j and Supplementary Fig. 5a–m). While most of the MOL2 in the dorsal horns of the spinal cord were derived from E12.5 pre-OPCs at P20, the contribution from postnatally fate mapped OPCs reached comparable levels in adulthood (19.09 ± 4.95% and 23.35 ± 1.28% from the postnatally- and embryonically derived MOL2, respectively; Supplementary Fig. 5m–o). This suggests that postnatally labeled OPCs start differentiating into MOL2 later than embryonically derived pre-OPCs.

**TFEB does not regulate the generation of MOL2 population.** Our scRNAseq and lineage tracing data suggest that the domain and time of developmental origin does not influence the OPC specification towards distinct MOL populations. Therefore, we assessed whether other intrinsic mechanisms might drive the region specificity of MOL subtypes. Recently, it has been shown that the transcription factor EB (TFEB), highly expressed in premyelinating oligodendrocytes[35], cell-autonomously regulates programmed cell death in the OL lineage and as such TFEB spatiotemporally controls the CNS myelination during development[36]. We conditionally knock-out TFEB taking advantage of the Olig2::Cre-TFEB mouse model[36]. We collected juvenile spinal cords from control and conditional KO (cKO) littermates and performed RNAscope ISH to detect the OL lineage (Sox10[+] cells), mature OLs (Aspa[+] cells)[5], MOL2 (Klk6[+]-Aspa[+] cells), and MOL5/6 (Ptgds[+]- Aspa[+] cells) in the gray and white matter of the spinal cord. We observed a consistent contribution of the mature OLs (Aspa[+]-Sox10[+] cells) to the lineage (Sox10[+] cells) in both genotypes (Supplementary Fig. 6a–c) in the spinal cord gray (Supplementary Fig. 6b) and white matter (Supplementary Fig. 6c). We also observed that all the Aspa[+] cells express Sox10, confirming that Aspa is a marker gene specific for the OL lineage[5].

When we analyzed the effect of the deletion of TFEB on the MOL2 and MOL5/6 populations, the number of Aspa[+] cells, MOL2 and MOL5/6 suggest a different representation of MOLs in the mutant mice (Fig. 4e). This was particularly evident in the gray matter of the Olig2::Cre[+]-TFEB[fl/fl] spinal cord tissue. Indeed, here we observed a low percentage of MOL5/6 and their more prominent reduction compared to Aspa[+] cells (Fig. 4a, b, d, e). To note, the GM of the spinal cord is a region where MOL5/6 are very abundant in WT (Fig. 2g, h) as well as in Olig2::Cre[−]-TFEB[fl/fl] control littermate mice (Fig. 3a, b, d). The percentage of the MOL5/6 calculated over the Aspa[+] cells in

the WM of Olig2::Cre[+]-TFEB[fl/fl] and Olig2::Cre[−]-TFEB[fl/fl] spinal cord did not show statistically significant changes (Fig. 4d, e). Similarly, the percentage of the MOL2 over the Aspa[+] cells in GM and WM of the spinal cord was comparable between genotypes (Fig. 4a–c, e). Our results suggest that TFEB-dependent programmed cell death of premyelinating OLs does not have a regulatory role in the specification of MOL populations in the spinal cord.

To further assess whether brain- and spinal cord-derived OPCs are intrinsically programmed to differentiate into MOL2 or MOL5/6, we also used an in vitro assay, the microfiber assay, as previously described[11,37]. The microfibers mimic an inert axonal net (3D environment); therefore, this assay has been used to evaluate the intrinsic potential of OPCs to generate OLs with different morphological features[11,37]. Here, we used it to determine the OPC intrinsic differentiation potential. We collected brains and spinal cords from P4-6 pups, dissociated the tissue and MACS-sorted for OPCs to seed on microfibers with a diameter range of 2–4 μm. We differentiated brain and spinal cord-derived OPCs separately for 3, 7, and 14 days in vitro and visualized the OL lineage and MOL-specific populations by ICC and RNAscope ISH to detect MBP and PDGFRα, as well as Itrp2, Dusp1, Klk6 or Ptgds, respectively. MBP[+] cells constitute approximately 10–20% of the cells cultured on the microfibers (Fig. 4f–h) at 3, 7, and 14 days of in vitro differentiation with no differences in the percentage of MBP[+] cells between timepoints in brain (p value 0.185) and spinal cord cultures (p value 0.8074) (Fig. 4f–h), Additionally, the morphological complexity of MBP[+] cells increases along time, suggesting that MBP[+] cells are differentiating into mature OLs (Fig. 4f, g, j, k). The time-course in brain-derived cultures showed that PDGFRα[+] OPCs were approximately 15% of the cells at 3 days and decrease considerably (p value 0.0363) at 14 days (Supplementary Fig. 6d, g). We also observed that 40% of the cells in brain-derived cultures at 3 days were newly formed OLs (NFOLs, Itrp2[+]), with a significant increase to nearly 60% (p value 0.019) at 14 days (Supplementary Fig. 6f, g). 40–60% of the MBP[+] cells were Itrp2[+] (Supplementary Fig. 6f, h), indicating that a large proportion of OLs had not transitioned to an MOL state.

When we quantified the percentage of Dusp1[+]- MBP[+] (MOL1), Klk6[+]- MBP[+] (MOL2) and Ptgds[+]- MBP[+] (MOL5/6) cells wrapping the microfibers, we did not observe any Klk6[+]-MBP[+] cells in culture (Fig. 4f, g, i, Supplementary Fig. 6d, e, g, h). While we could detect rare Dusp1[+]-MBP[+] cells at 14 days (Supplementary Fig. 6e, h), the vast majority of Dusp1[+] were not MOLs, but most likely OPCs and VLMCs, consistent with the expression of Dusp1 also in these populations[5]. Instead, we observed around 40% Ptgds[+]- MBP[+] cells in both brain and spinal cord OL lineage cultures (Fig. 4f–i). These data suggest that OPCs mainly differentiate into NFOLs and MOL5/6, but not into MOL1 or MOL2 in standard culture conditions[11,37]. Altogether our in vivo and in vitro data suggest that OPCs are not intrinsically programmed to generate distinct MOL subpopulations. Therefore, more plausibly, the exposure of OPCs to extrinsic signals during critical windows of migration or differentiation during development might regulate the differentiation potential of OPCs.

**MOL2 and MOL5/6 have different responses to traumatic spinal cord injury.** The spatial preference of MOL2 and MOL5/6 for different regions and tracts of the spinal cord might relate to functional differences in disease. Traumatic injury of the spinal cord is a chronic pathological condition that leads to loss of locomotor and sensory functions largely due to Wallerian degeneration[38]. Following injury, myelin is lost at the injury site

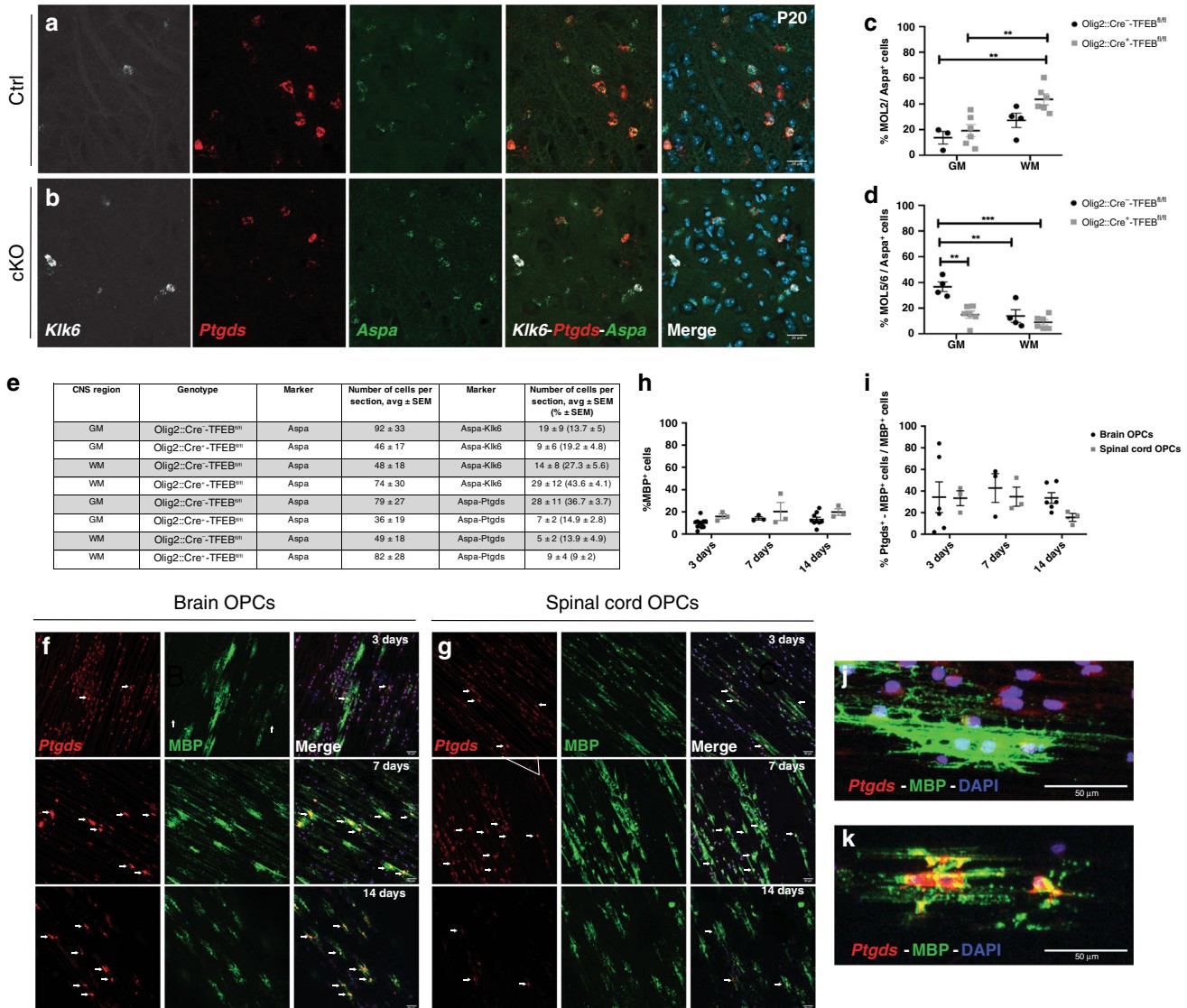

| CNS region | Genotype | Marker | Number of cells per section, avg ± SEM | Marker | Number of cells per section, avg ± SEM (% ± SEM) |
|---|---|---|---|---|---|
| GM | Olig2::Cre⁻-TFEBfl/fl | Aspa | 92 ± 33 | Aspa-Klk6 | 19 ± 9 (13.7 ± 5) |
| GM | Olig2::Cre⁺-TFEBfl/fl | Aspa | 46 ± 17 | Aspa-Klk6 | 9 ± 6 (19.2 ± 4.8) |
| WM | Olig2::Cre⁻-TFEBfl/fl | Aspa | 48 ± 18 | Aspa-Klk6 | 14 ± 8 (27.3 ± 5.6) |
| WM | Olig2::Cre⁺-TFEBfl/fl | Aspa | 74 ± 30 | Aspa-Klk6 | 29 ± 12 (43.6 ± 4.1) |
| GM | Olig2::Cre⁻-TFEBfl/fl | Aspa | 79 ± 27 | Aspa-Ptgds | 28 ± 11 (36.7 ± 3.7) |
| GM | Olig2::Cre⁺-TFEBfl/fl | Aspa | 36 ± 19 | Aspa-Ptgds | 7 ± 2 (14.9 ± 2.8) |
| WM | Olig2::Cre⁻-TFEBfl/fl | Aspa | 49 ± 18 | Aspa-Ptgds | 5 ± 2 (13.9 ± 4.9) |
| WM | Olig2::Cre⁺-TFEBfl/fl | Aspa | 82 ± 28 | Aspa-Ptgds | 9 ± 4 (9 ± 2) |

**Fig. 4 TFEB does not regulate OPC differentiation into distinct MOL subpopulations during development and the OL lineage heterogeneity observed in vivo is not modeled in 3D culture conditions. a, b** Confocal representative images show that MOL2 (Klk6⁺-Aspa⁺ cells) and MOL5/6 (Ptgds⁺-Aspa⁺ cells) in the juvenile (P20) gray matter of the spinal cord of the Olig2::Cre⁻-TFEBfl/fl (**a**) and Olig2::Cre⁺-TFEBfl/fl (**b**) mice. Scale bar = 20 μm. **c, e** Quantification of the MOL2 (Klk6⁺-Aspa⁺ cells, **c**), and MOL5/6 (Ptgds⁺-Aspa⁺ cells, **d**) populations in the gray and white matter of the juvenile (P20) spinal cord, and summary table (**e**). Percentages of the populations are calculated on the total number of mature OLs (Aspa⁺ cells) in the analyzed region. Data are presented as mean ± SEM. n = 3−6 animals per genotype, and values of the individual data points are reported in the Source Data file. Asterisks indicate a significant difference between conditions (**p ≤ 0.01, ***p ≤ 0.0001, two-way ANOVA with Sidak's correction). Exact p values are reported in the Source Data file. **f, g** Confocal representative images show the MBP⁺ and Ptgds⁺-MBP⁺ cells derived from brain (**f**) and spinal cord (**g**) OPCs cultured on microfibers for 3, 7, or 14 days of differentiation in vitro. Scale bar = 50 μm. **h, i** Quantification of the MBP⁺ (**h**) and Ptgds⁺/MBP⁺ (**i**) cells cultured on microfibers for 3, 7 or 14 days of differentiation in vitro. Percentages of the populations are calculated on the total number of nuclei (**h**) or MBP⁺ cells (**i**). Data are presented as mean ± SEM. n = 3−11 independent experiments. Each experiment was a biological replicate. Tissue from n = 4−10 P7 pups was used for each biological replicate. OPC oligodendrocyte progenitor cell, Ctrl Olig2::Cre⁻-TFEBfl/fl, cKO Olig2::Cre⁺-TFEBfl/fl, GM gray matter, WM white matter. **j, k** Representative images of MBP⁺ cell from brain OPCs 3 days in vitro with process extension on top of microfibers (**j**) and 14 days in vitro with some processes enwrapping microfibers (**k**). OPC oligodendrocyte progenitor cell. Source data are provided as a Source Data file.

and partially around the axons spared by the injury. We took advantage of the dorsal funiculi transection, a mild model of traumatic injury (Fig. 5a, b). We observed that the OL lineage is well represented distal to and at the injury site, both during acute (14 days post-injury, Fig. 5d) and chronic (3- and 5 months post-injury, Fig. 5f, h, i) phases following injury[39–43]. During the acute phase following injury, when part of the OL lineage undergoes cell death, we observed that MOL2 and MOL5/6 contribute to the OL lineage at the injury site in a similar manner (Fig. 5c, e). In contrast, during the chronic phase (when remyelination occurs),

we observed a decreased contribution of MOL2 to the OL lineage at the injury site (Fig. 5f, g, j, k). While MOL5/6 reached a higher contribution to the OL lineage (Fig. 5f, g, j, k) compared to the naive spinal cord (Fig. 2), despite the absence of intact axons in the injury site.

We also assessed the spatial distribution of MOL2 and MOL5/6 in regions of Wallerian degeneration rostral and caudal to the injury site (Fig. 5f, g, j, k and Supplementary Fig. 7a–e). We observed that the spatial preference of MOL2 and MOL5/6 for the dorsal columns and dorsal corticospinal tract was unaffected by

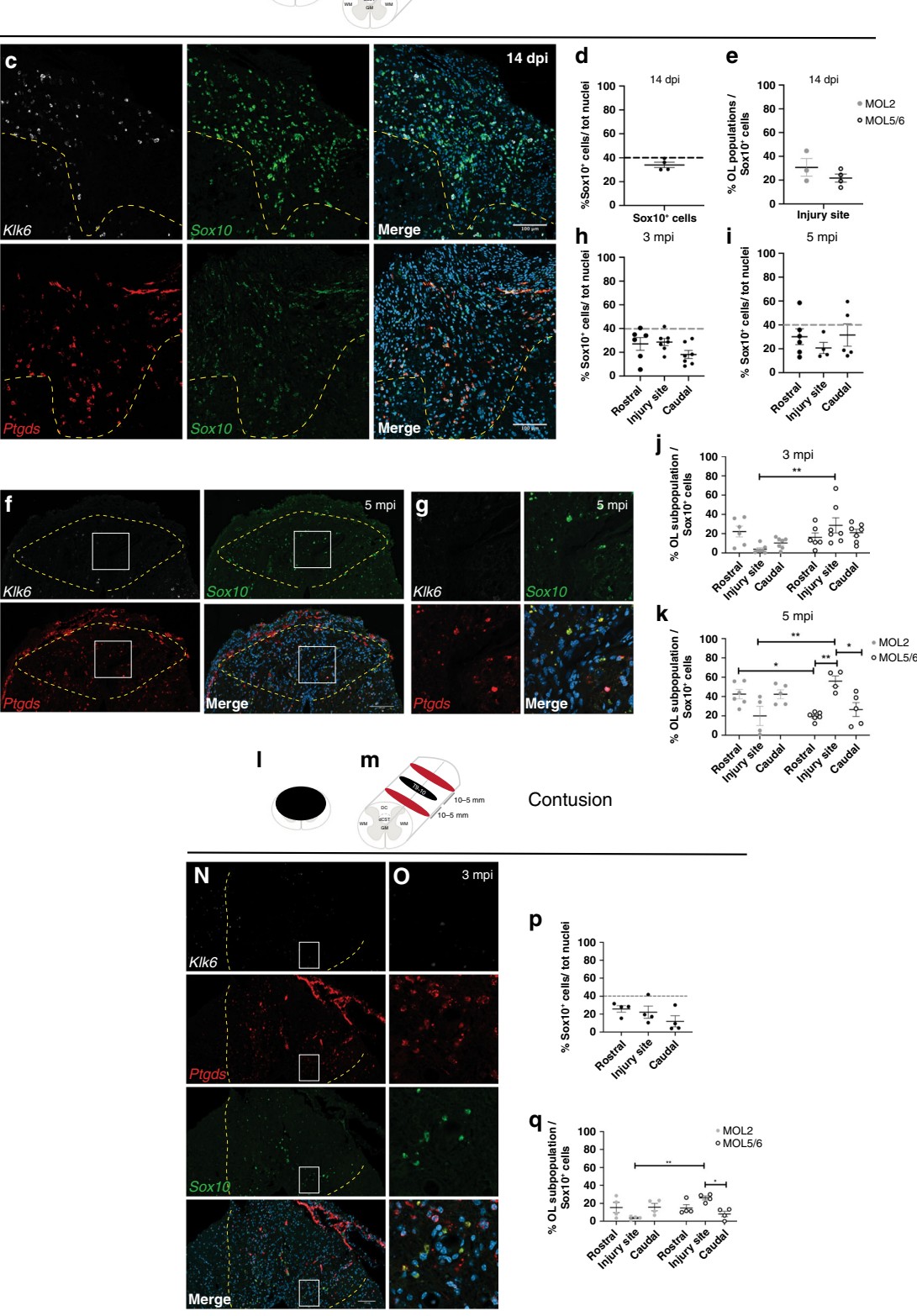

Dorsal funiculi transection

Contusion

the injury (Supplementary Fig. 7a–e). The contribution of MOL5/6 in the dorsal funiculi rostral and caudal to the injury did not change following injury (18.8 ± 2.82% and 18.75 ± 2.9% at 3 and 5 mpi, respectively; Fig. 5j, k and Supplementary Fig. 7a–e), and remained similar to their contribution in the white matter of the intact adult spinal cord (Fig. 2 and Supplementary Data 3).

Surprisingly, the contribution of MOL2 to the OL lineage in the same regions increased with time (Fig. 5j, k). Indeed, 5 mpi, the MOL2 cell density in WD is even greater than found in the WM of the intact adult spinal cord (Fig. 2 and Supplementary Data 3). We also investigated the dynamics of MOLs following severe contusion, another model of traumatic injury (Fig. 5l, m). As in

**Fig. 5 MOL2 and MOL5/6 show differential susceptibility to disease. a**, **b**, **l**–**m** Schematic of the different models of traumatic spinal cord injury as well as the lesions relative extent and distribution (**a**, **l**), level of the lesions (T9-10. **b**, **m**) and distance of the rostral and caudal analyzed segments (**b**, **m**). **c**, **f**, **g**, **n**, **o** Confocal representative images of the lesions following dorsal funiculi transection (**c**, **f**, **g**) and contusion (**n**, **o**) injuries, showing the specific loss of MOL2 (*Klk6*+ OL lineage cells) and the high repopulation by MOL5/6 (*Ptgds*+ OL lineage cells) of the lesions during the chronic phase following traumatic spinal cord injury (**f**, **g**, **o**), but not at the acute phase (**c**). Yellow dashed lines highlight the lesion sites. White rectangles highlight the regions shown in higher magnification. Scale bar = 100 μm. **d**, **e**, **h**, **i**, **j**, **k**, **p**, **q** Quantification of the OL lineage (**d**, **h**, **i**, **p**) and the MOL populations away and at the injury sites (**e**, **j**, **k**, **q**). Percentage of the MOL2 and MOL5/6 subpopulations was calculated on the total number of OL lineage cells (*Sox10*+ cells). Dashed black line marks the average percentage of the OL lineage cells in the intact adult spinal cord (**d**, **h**, **p**). Data are presented as mean ± SEM. $n = 3$–7 animals per condition, and can be assessed in the Source Data file. Asterisks indicate a significant difference between conditions (*$p \leq 0.05$, **$p \leq 0.01$, two-way ANOVA with Sidak's correction). Exact $p$ values are reported in the Source Data file. SCI spinal cord injury, dpi days post-injury, mpi months post-injury, T10 thoracic vertebra 10. MOL mature oligodendrocyte. Source data are provided as a Source Data file.

the dorsal funiculi transection model, we observed that the OL lineage is well represented during the chronic stage (3 months post-injury; Fig. 5n, p), as well as a decreased contribution of MOL2 to the OL lineage at the injury site, and increased representation of MOL5/6 (Fig. 5n, o, q), confirming that MOL2 and MOL5/6 populations respond differently to traumatic injury during the remyelination phase.

**MOLs change their transcriptional profile following traumatic spinal cord injury.** To investigate the molecular changes in the OL lineage following traumatic injury, we collected the injury site and regions of Wallerian degeneration rostral and caudal to injury as well as thoracic spinal cord segments from laminectomy control Sox10::Cre-GFP mice, FACS sorted the OL lineage, and performed scRNAseq. Graph-based clustering (Seurat)[27,28] and integration with the Marques et al.[5] and Zeisel et al.[14] scRNA-Seq datasets (Fig. 6a, b and Supplementary Fig. 8a–c, g) allowed the identification of rare cells types as ependymal cells (positive for *Dynlrb2*, *Tmem212* and *Ccdc153*)[14] plausibly SCI-activated ependymal cells differentiating into OL lineage cells, immune cells (some of which with myelin transcripts, thus most likely phagocyting microglia/macrophages), and Schwann cells (Supplementary Fig. 8a–c), which have been previously shown to arise from OPCs in the injury site upon demyelination[44] and SCI[45]. Frequency analysis highlighted that MOL1 (and MOL5, although to a lesser extent) are enriched in the injury site when compared to regions of Wallerian degeneration and control (Fig. 6c, d). In contrast, MOL2, but also MOL6, are depleted at the injury site (Fig. 6c, d). The MOL2 population decreased its contribution to the lineage, but it does not appear to be as dramatically lost at the injury site as detected by RNAscope ISH (Fig. 5f, g, j, k, n–q). This partial discrepancy could be due to the dissection of the injury area for scRNAseq. Indeed, dissection included the whole lesion site and a rim of spared tissue around the lesion due to technical challenges, while the RNAscope ISH analysis has been performed on tissue sections of the injury epicenter, where the lesion area is the largest. More importantly, differential gene expression analysis comparing the OL lineage at the injury site and the laminectomy control SCI or regions of Wallerian degeneration above and beyond injury against control showed that gene markers of the MOL2 population (*Hopx*, *Spock3*) are downregulated in the injury site, while MOL1 marker genes, such as *Egr1*, *Fos*, *Fosb*, are enriched in the injury site and decreased in regions of Wallerian degeneration (Fig. 6e, f). Signature marker genes of the MOL5/6 population are expressed at comparable expression levels (*Ptgds*, *Jph4*, *Npsr1*) (Fig. 6e, f).

To assess the molecular mechanisms that change in the OL lineage and potential different roles of distinct MOL populations following SCI, we identified the differentially expressed genes in the distinct populations (Supplementary Data 5) and performed gene ontology and pathway analysis (Supplementary Data 6 and 7). The differential gene expression analysis is performed on the

average gene expression levels of the cells within each sample. For the gene ontology and pathway analysis, we firstly compared the differentially expressed genes of each MOL population between laminectomy control and injury site. We identified the upregulation of genes associated with cholesterol biosynthesis and steroid metabolism (*Msmo1*, *Sqle*, *Hmgcs1*, *Idi1*) in MOL5 and antigen presentation (*B2m*, *H2-D1*) and as well as platelet degranulation, activation, and aggregation (*Psap*, *Anxa5*, *Cd63*, *Sparc*) in MOL6 at the injury site (Supplementary Data 6). On the other hand, we identified downregulation of genes associated with Class I MHC-mediated antigen processing and cross-presentation (*Psme1*, *Psmb10*, *Tap1*, *Psmb9*, *Tap2*, *Psmb8*) in the MOL1 population, transmission across chemical synapses (*Gng11*, *Calm1*, *Calm2*, *Sdc38a2*), and RHO GTPase signaling (*Calm1*, *Calm2*, *Ppp1r14a*, *Tubb4a*) in the MOL6 population at the injury site compared to laminectomy control (Supplementary Data 7).

We also observed several changes when comparing the MOL1-6 populations in the injury site and regions of Wallerian degeneration (Supplementary Data 6 and 7). Briefly, MOL2, MOL5 and MOL6 upregulate genes associated with endosomal, ER-phagosome, and vacuolar pathway and immunoregulatory interaction between lymphoid and non-lymphoid cells (*B2m*, *H2-K1*, *H2-D1*) in the injury site compared to regions of Wallerian degeneration. MOL5 also upregulate genes associated with cholesterol biosynthesis and steroid metabolism (*Msmo1*, *Sqle*, *Hmgcs1*, *Idi1*). Platelet degranulation, activation, and aggregation (*Psap*, *Anxa5*, *Cd63*, *Sparc*) are upregulated in MOL6, which downregulate genes associated with the RHO GTPase signaling (*Calm1*, *Calm2*, *Ppp1r14a*, *Tubb4a*). MOL1 downregulate genes associated with plasma lipoprotein assembly (*Abca1*, *Apoe*) and ion homeostasis/transport (*Fxyd7*, *Fxyd1*). When comparing the MOL populations in the laminectomy control and regions of Wallerian degeneration, we did not observe any robustly upregulated pathway in the OL lineage, but we observed that MOL5 in the regions of Wallerian degeneration downregulate genes associated with endosomal, ER-phagosome, and vacuolar pathway and immunoregulatory interaction between lymphoid and non-lymphoid cells (*B2m*, *H2-K1*, *H2-D1*) compared to the MOL5 population in the laminectomy control. Despite some consistently differentially expressed genes among populations (e. g, *Calm1*, *Calm2*), our gene ontology and pathway analysis show that the MOL populations undergo population-specific transcriptional changes that might result in MOL population-specific functions following traumatic injury (Fig. 6f and Supplementary Fig. 8d–f).

**Decrease in MOL2 is not a general signature of demyelination and axonal pathology.** In a previous study, we took advantage of EAE[7], a mouse model of CNS inflammation presenting multifocal lesions characterized by extensive oligodendrocyte loss and Wallerian degeneration. We induced EAE by immunizing with MOG$_{35-55}$ transgenic mouse lines on a C57B6/J background[7].

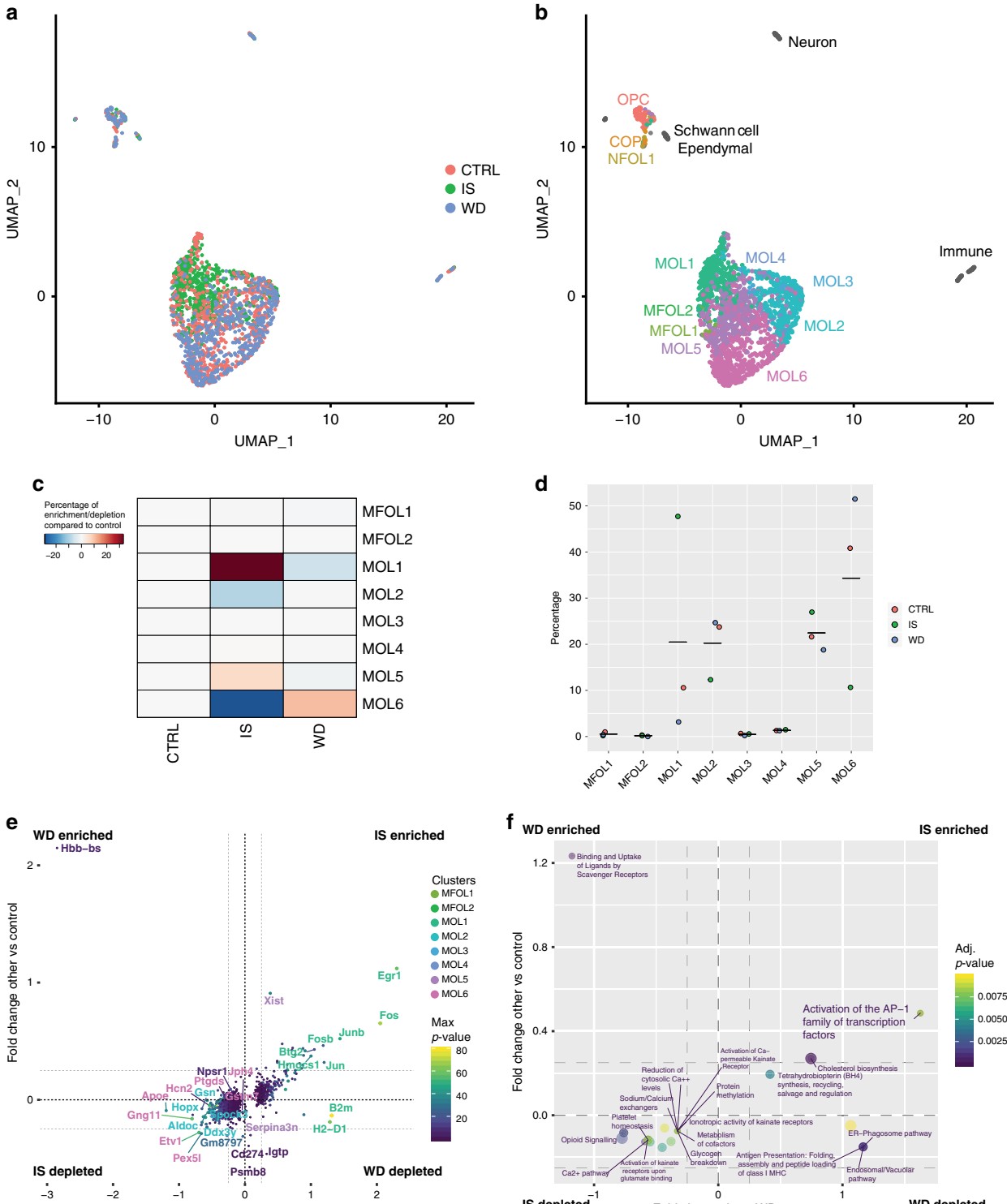

**Fig. 6 MOL2 and MOL5/6 change their transcriptional profile following traumatic spinal cord injury. a, b** Uniform Manifold Approximation and Projection (UMAP) plots showing 1967 Sox10::Cre-GFP-positive cells from laminectomy control, injury site, and Wallerian degeneration regions contributing to the clusters (**a**) and cell lineage composition (**b**), as determined by graph-based clustering (Seurat)[27,28] and integration of labels with the Marques et al. scRNA-Seq dataset[5,17]. **c, d** Frequency distribution of the OL lineage cells from the laminectomy control and injury site (**c**) and Wallerian degeneration regions and injury site (**d**) forming the clusters. **e, f** 2D Volcano plots showing the differentially expressed genes (**e**) and biological processes (**f**) in the OL lineage cells from the injury site compared to laminectomy control and Wallerian degeneration regions. Gene symbols are colored according to the clusters that are enriched at, while the corresponding dot is colored according to the p value. The OL lineage was sorted from Sox10::Cre-GFP mouse line 3 months following spinal cord injury. CTRL laminectomy control, IS injury site, WD Wallerian degeneration. OPC oligodendrocyte progenitor cell, COP committed OPC, NFOL newly formed oligodendrocyte, MFOL myelin forming oligodendrocyte, MOL mature oligodendrocyte. Source data are provided as a Source Data file.

This model is severe and allows analysis of EAE at the peak of disease, but it does not present remitting-relapsing phase. We showed that the OL lineage presents some EAE-associated states of MOL2 and MOL5/6 that are characterized by the expression of unique EAE-induced genes in addition to the signature genes for the population[7]. Importantly, those data show that the MOL2 population does not downregulate the expression of *Klk6* in the disease context (Supplementary Fig. 9i), suggesting that *Klk6* gene expression is not regulated by disease but it is a stable MOL2 marker.

In the EAE model, the OL lineage cells are mostly absent at the lesion site and reduced of approximately 50% in the peri-lesion areas compared to the spinal cord white matter of control conditions (*p* value 0.0410 and 0.1000 for total number of cells) (Supplementary Fig. 9a–f). We used this model of demyelination to assess the spatial distribution of the MOL2 and MOL5/6 in the peri-lesion area at the peak of disease (Supplementary Fig. 9g, h). We observed a similar distribution of MOL2 and MOL5/6 in the intact or lesioned white matter of the spinal cord in mice immunized with CFA or MOG$_{35-55}$, respectively (Supplementary Fig. 9g, h), consistently with our previous transcriptomic analysis, which showed that the MOL2 population is not lost in the spinal cord of EAE mice[7].

In summary, our data indicate that MOL2 decrease their contribution to the OL lineage at the injury site and increase their contribution to the OL lineage in regions of Wallerian degeneration specifically during the chronic phase following traumatic SCI. These data suggest that MOL2 may be associated with circuit remodeling, resulting from sprouting of intact axons proximal to the injury site. On the other hand, the increased contribution of MOL5/6 to the OL lineage at the injury site suggests that the injury site may present factors that stimulate the resident OPCs to preferentially differentiate into MOL5/6 [4,46,47]. Altogether, our data suggest that MOL2 and MOL5/6 respond differently to SCI and that they might have different roles during regenerative phases in disease.

## Discussion

Functional heterogeneity of MOLs might correlate with the local environment or their interaction with different neuron types[48]. Here, we unveiled that distinct MOL populations have spatial preference in the mammalian central nervous system and differential susceptibility to traumatic SCI. Additionally, we also show that the generation of distinct MOL populations is independent of the OPC intrinsic mechanisms, such as their time and domain of origin during development or programmed cell death[29,30,36]. Indeed, our observations fit a model where the mechanisms regulating the generation of the OL lineage are independent of the OPC developmental origin or developmental selection driven by programmed cell death[25,29,31], differently than it is for subset of neuronal populations such as cortical interneurons[49–52], but with certain similarities to what has been observed with post-mitotic layer 4 spiny neurons[53].

Our data suggest that MOL5/6 might be an OL population associated with adaptive myelination, as MOL5/6 increases in the corpus callosum and CST from juvenile to adulthood. Indeed, the tracts forming both the corpus callosum and the CST maintain plasticity throughout life, essential for acquisition and maintenance of new motor skills with continuous OL turnover, suggesting that myelin needs to be constantly remodeled (adaptive myelination) in those regions[23,24,33]. In contrast, we showed that MOL2 preferentially locate in the fasciculi gracilis and cuneatus (sensory tracts of the spinal cord), suggesting that MOL2 might be associated with more stable neural circuits or production of myelin favoring fast conduction. Indeed, the proprioceptive

and mechanoreceptive long projecting fast axons forming the ascending tracts in the dorsal funiculi are myelinated early in development and reach complete myelination by the juvenile stage[24,54], correlating with the early, high and stable enrichment of MOL2 in this region.

Here, we also report that the MOL2 population is particularly affected by traumatic injury, as this population is lost at the injury site and it does not repopulate this region even 5 months after injury, a time point when the physiological partial remyelination and repair has occurred. Furthermore, we observed that MOL2 and MOL5/6 are still quite abundant in regions where the axons forming the dorsal funiculi are lost, as they underwent Wallerian degeneration (Supplementary Fig. 7d, e). The presence of so-called quiescent oligodendrocytes has been previously described in Wallerian degeneration[55,56]. Here we show that axonal degeneration does not affect the relative contribution of MOL2 and MOL5/6 to the OL lineage, suggesting that MOLs have stable identities although their surrounding axons are degenerating or lost. Our results show that specific mature OL populations present different response to traumatic injury. The gene ontology and pathway analysis suggest that MOL2 and MOL5/6, as well as other MOL populations, might have different functions, which might underlie MOL population-specific contributions to remyelination, axonal support, action potential conduction, and synapse formation inhibition or other phenomena. Distinct MOL populations might differentially contribute to neuroregeneration and repair following SCI. We anticipate that our study will pave the way for further understanding of the MOL population-specific functional roles in development, health, and disease, allowing for better targeting of the OL subtypes important for the regeneration and repair of the central nervous system.

## Methods

**Animals.** All experimental procedures in this study were conducted in accordance with the European directive 2010/63/EU, local Swedish directive L150/SJVFS/2019:9, Saknr L150 and Karolinska Institutet complementary guidelines for procurement and use of laboratory animals, Dnr 1937/03-640. The procedures described here were approved by Stockholms Norra Djurförsöksetiska nämnd, the local committee for ethical experiments on laboratory animals in Sweden, lic.nr. 130/15, 144/16, and 1995/2019.

Mouse lines used in this study are Pdgfrα::CreER$^{TM}$-RCE::LoxP- RCE (Z/EG, mixed C57BL/6NJ and CD1 background) (The Jackson Laboratory, stock nr. 018280)[30], Sox10::CreER$^{T2}$-ROSA26::LoxP-GFP (C57B6/J, http://www.informatics.jax.org/allele/MGI:5301107, from Vasillis Pachnis, Francis Crick Institute, and William Richardson, University College of London)[57,58], Emx1::Cre-Sox10::Cre-LoxP-GFP-STOP-TdTom (from William Richardson, University College of London, mixed CBA and C57BL/6 background)[26] and Olig2::Cre$^+$-TFEB$^{fl/fl}$[36]. Mice were used with the Cre allele in hemizygosity and the reporter gene allele in either hemizygosity or homozygosity. All animals were free from the most common mouse viral pathogens, ectoparasites, endoparasites, and mouse bacteria pathogens harbored in research animals. The battery of screened infective agents met the standard health profile established in Karolinska Institutet animal housing facilities. For all the mouse strains, we used animals of both sexes at P20 or P60.

*Animal husbandry.* Mice were housed to a maximum number of 5 per cage in individually ventilated cages (IVC sealsafe GM500, Tecniplast). Cages contained hardwood bedding (TAPVEI, Estonia), nesting material, shredded paper, gnawing sticks and card box shelter (Scanbur). The mice received regular chew diet (either R70 diet or R34, Lantmännen Lantbruk, Sweden). General housing parameters such as relative humidity, temperature, and ventilation follow the European convention for the protection of vertebrate animals used for experimental and other scientific purposes treaty ETS 123. Briefly, consistent relative air humidity of 50%, 22 °C and the air quality is controlled with the use of stand-alone air handling units supplemented with HEPA filtrated air. Monitoring of husbandry parameters is done using ScanClime (Scanbur) units. Water was provided by using a water bottle, which was changed weekly. Cages were changed every other week. All cage changes were done in a laminar air-flow cabinet. Facility personnel wore dedicated scrubs, socks and shoes. Respiratory masks were used when working outside of the laminar air-flow cabinet. Animals were sacrificed at juvenile (P20−21) and adult stages (P60), injury of the spinal cord was performed on adult mice (P60−P150) and sacrificed 14 days, 3- or 5 months post-injury, EAE was induced in adult mice (P90) and terminated when the MOG35-55 (EK-2110, Hooke laboratories) immunized mice reached a clinical score of 3.0 (limp tail and almost complete or

complete paralysis of hind legs. P100−110). Both sexes were included in the experiments, except for the contusion SCI paradigm, where only female mice were used. The following light/dark cycle was used: dawn 6:00−7:00, daylight 7:00−18:00, dusk 18:00−19:00, night 19:00−6:00.

**Lineage tracing**. To fate map the OPCs generated at E13.5 and present at P3−5, Pdgfrα::CreER$^{TM}$-RCE::LoxP- RCE (Z/EG) mice were used[30]. Time-mated females were injected i.p. with 1 mg of tamoxifen (T5648, Sigma) at pregnancy day E12.5 or 2 mg once daily when pups were P3−P5 (tamoxifen reaching the pups via the mother's milk). Low dose (1 mg) of tamoxifen during pregnancy (equivalent to 33 mg/kg of body weight) was used to restrict the labeling to the first appearing OPCs. Indeed, at this low dose, tamoxifen is metabolized within 24−36 h after injections (therefore E13.5−E14.0)[59,60]. The litters were then sacrificed as juvenile (P20) or young adults (P60) and brains and spinal cords collected for tissue analysis by RNAscope ISH coupled with IHC.

**Spinal cord injury and postoperative care**
*Dorsal funiculi transection: mild injury*. Mice were kept under anesthesia with a 2% isoflurane (IsoFlo®, vet 100%)/O$_2$ mixture at 1 l/min, after 2−3 min induction with 5% isoflurane/O$_2$ mixture at 1 l/min. Body temperature was maintained by keeping the mice on a heating pad (37−39 °C) during the whole procedure.

The injury site was mid-thoracic (T10). The fur was shaved and the area disinfected with chlorhexidine alcohol (Fresenius Kabi) and 70% EtOH (once). The skin was incised, the superficial fat displaced, the prominent vessel (between T7 and T8) was identified and used as reference point. Then, the muscle tissue over T9 −11 removed to expose the vertebrae. A T10 laminectomy was performed, the dura mater was removed, and the dorsal funiculi transection was performed with a microknife (5 mm, 15°Stab, Sharpoint). Dorsal funiculi transection damages dorsal columns, dorsal corticospinal tract, and partially the dorsal horns. After surgery, the mice were injected i.p. with Buprenorphine (Temgesic®) 0.01 mg/kg of body weight and s.c. with 0.5 ml of 0.9% saline solution. The mice were then placed in their home cages and monitored until fully recovered from anesthesia. During their postoperative care, mice underwent daily checks for general health status, weight loss, mobility, wounds, swelling, infections, or autophagy of the toes. When mice lost weight after surgery, their diet was supplemented with DietGel® Recovery (72-06-5022, Clear H$_2$O). The mice used in this study did not show self-induced wounds or autophagy of the toes or wound infections.

*Contusion: severe injury*. Mice were deeply anesthetized with isoflurane and provided with pre-operative analgesia (Buprenorphine, Schering-Plough, 0.1 mg/kg body weight and Carprofen, Pfizer, 5 mg/kg body weight). A laminectomy was performed at the T8−T9 level to expose the dorsal portion of the spinal cord. The vertebral column was then stabilized with clamps (Precision System Instru-mentation, PSI) placed on the foramen of T8 and T10. After the administration of local anesthesia (Xylocaine/Lidocaine, AstraZeneca, 10 mg/ml, two drops on the spinal cord surface), a 70 kDyne contusion was delivered medially over T9 using the Infinite Horizon Impactor (IH-0400, PSI) equipped with a 1.3 mm tip. The muscle wall and skin were then sutured (Vicryl plus, 70 cm M2, VCPV327H & Ethilon, 4-0, FS-2, 45 cm, EH7144H respectively), and mice placed in a heated cage until they regained consciousness after which they were transferred to their home cage that was equipped with an elevated floor grid. During the first 3 days of postoperative care, animals received additional hydration (500 μl saline injection daily), antibiotic treatment (Sulphadizine/Trimethoprim, Tribrissen vet., MSD, 100 mg/kg body weight per day) and analgesia (Buprenorphine, Schering-Plough, 0.1 mg/kg body weight and Carprofen, Pfizer, 5 mg/kg body weight per day). Bladders were expressed three times per day during the first 3 days and twice per day until they regained bladder control. Their diet was supplemented with a high energy nutritional supplement (DietGel Boost, Clear H$_2$O) during the first week and starting from 1 to 2 days prior to the surgery. Body weight was mon-itored daily during the first week and weekly thereafter. Animals that lost more than 15% of their pre-operative body weight were euthanized and not included in the study.

**Experimental autoimmune encephalomyelitis**. For the induction of chronic EAE, animals on C57B6/J genetic background were injected subcutaneously with an emulsion of 150 ng MOG35-55 (1 mg antigen/ml) in CFA or vehicle (EK-2110, Hooke Laboratories), followed by the i.p. administration of pertussis toxin (0.2 μg per animal, EK-2110, Hooke Laboratories) in phosphate-buffered saline 1× (10010023, Gibco) for 2 consecutive days (accordingly to the manufacturer's instructions). All animals were monitored daily twice morning and afternoon to assess general health status and clinical signs of EAE disease. When mice lost weight after surgery, their diet was supplemented with DietGel® Recovery (72-06-5022, Clear H$_2$O). One mouse showed signs of infection at the subcutaneous injection site and was humanely terminated. The mice were also daily score for clinical sign of EAE. Briefly, 0 = normal, no obvious changes in motor function; 0.5 = Tip of tail is limp, tail moves normally; 1.0 = limp tail, no signs of tail movement are observed; 1.5 = limp tail and hind leg inhibition; 2.0 = limp tail and weakness of hind legs, poor body balance; 2.5 = limp tail and dragging of hind legs; 3.0 = limp tail and almost complete or complete paralysis of hind legs; 3.5 = limp

tail and complete paralysis of hind legs, when the animal is placed on its side, is unable to right itself; 4.0 = limp tail, complete hind leg and partial front leg paralysis, animal is minimally moving. At score 3.0 animals were euthanized.

**Tissue preparation and sectioning**. At the end of the experiments for the visualization in tissue of the OL lineage populations, the animals were deeply anesthetized with ketamine (120 mg/kg of body weight) and xylazine (14 mg/kg of body weight) and transcardially perfused with 0.1 M Phosphate Buffered Sal-ine (PBS) pH 7.4 (10010023, Thermofisher) followed by 4% acid-free pH 7 phosphate-buffered formaldehyde solution (Carl Roth). Brains and spinal cords were dissected and post-fixed in 4% paraformaldehyde (PFA) in PBS (pH 7.4) at 4 °C overnight and cryoprotected in 30% sucrose in PBS pH 7.4 for 48−36 h.

Tissue was embedded in Tissue-Tek® O.C.T. compound (Sakura). Both brains and spinal cords were coronally cryosectioned (20 and 16 μm, respectively) in 1:10 series. Sections were stored at −80 °C until further use.

**Tissue dissociation and single-cell RNAseq**. The corpus callosum of P60 Emx1::Cre-Sox10::Cre-LoxP-GFP-STOP-TdTom[26] was sectioned at the microtome and microdissected from two mice. The injury site, rostral and caudal Wallerian degeneration regions or control spinal cord of 3 months Sox10::Cre-GFP mice were microdissected from eight injured and five laminectomy mice. The dissected tissue was pooled and dissociated into a single-cell suspension, as previously described[5]. Briefly, mice were transcardially perfused with ice-cold oxygenated artificial cere-brospinal fluid (22 mM NaCl, 0.63 mM KCl, 0.4 mM NaH$_2$PO$_4$ * 2H$_2$O, 6.5 mM NaHCO$_3$, 25 mM Saccharose, 5 mM Glucose, 0.5 mM CaCl$_2$, 4 mM MgSO$_4$. pH 7.3) and the brains and spinal cord collected. Tissue dissociation was performed with the Adult Brain Dissociation Kit (Miltenyi Biotec) following the manu-facturer's instructions (red blood cells removal step was not included). The myelin debris removal was performed on the corpus callosum samples only. Lastly, the cells were resuspended in ice-cold 1% Bovine Serum Albumin (BSA) in artificial cerebrospinal fluid, filtered with 30 μm filter (Sysmex Partec) and FACS sorted with the BD Influx System (USB. BD FACS™) to collect GFP$^+$ and TdTom$^+$ OL lineage cells.

The sorted cells were processed with the Chromium Single Cell A Chip kit v2 (corpsus callosum) and v3 (spinal cord) and library prep with the Chromium Single Cell 3′Library & Gel Beads kit v2 (corpus callosum) and v3 (spinal cord. 10× Genomics) accordingly to the manufacturer's instructions. A total of 3000 cells for each sample was loaded on the Chromium Single Cell A Chip, although a lower number of cells was recovered in singlet and passed the quality control. The scRNAseq datasets is available at GEO (NCBI) at GSE128525. A web resource for browsing differential gene expression data for the single-cell data can be accessed at https://ki.se/en/mbb/oligointernode.

**RNAscope in situ hybridization (ISH), immunostaining (IHC), and immuno-cytochemistry (ICC)**. RNAscope ISH was performed using the RNAscope® Multiplex Fluorescent Detection Reagents Kit v2 (ACD Biotechne) on PFA fixed juvenile, adult, and injured brains and spinal cords according to the manufacturer's instructions with some modifications. Briefly, after treatment with boiling 1× target retrieval, the sections were incubated with Protease IV for 20 min at RT, followed by washing and the indicated hybridization and amplification steps. Probes used in this study were designed for mouse Sox10-C1 or -C2 (ACD Biotechne, 435931), Ptgds-C1 (ACD Biotechne, 492781), Klk6-C3 (ACD Biotechne, 493751), Egr2-C3 (ACD Biotechne, 407871), Ptprz1-C1 (ACD Biotechne, 460991), Itpr2-C1 (ACD Biotechne, 462071), and Dusp1-C1 (ACD Biotechne, 424501).

For lineage tracing experiments and the identification of P20 and P60 OL lineage within tissue from the Pdgfrα::CreER$^{TM}$-RCE::LoxP-GFP mice[30,61], the RNAscope ISH was coupled with IHC to detect the GFP reporter or endogenous Sox10. Briefly, after hybridization and amplification steps to detect the mRNA of the target gene markers, the sections were blocked in 5% normal donkey serum (NDS), 0.03% Triton X100 in PBS for 1 h at RT and incubated with chicken anti-GFP (AbCam, ab 13970) or goat anti-Sox10 (Santa Cruz, sc-17342) primary antibodies 1:200 in 2% NDS, 0.03% Triton X100 in PBS, O.N. at RT. The following day, the sections were incubated with goat anti-chicken AlexaFluor 488 conjugated (AbCam, ab150169) or donkey anti-goat AlexaFluor 647 conjugated (LifeTech, A21447) secondary antibodies 1:500 in 2% NDS, 0.03% Triton X100 in PBS, 1 h at RT and counterstained with DAPI (1:5000 in PBS) for 2 min. IHC washing steps were performed with 0.05% Tween-20 in PBS.

To stain cells on microfibers, medium was removed, and cells were washed once with 1× PBS and then fixed in 4% PFA for 15 min at RT. Cells were stained by coupling RNAscope with immunocytochemistry. Briefly, cells were treated with diluted Protease III (1:15 in water) for 10 min at RT followed by washing and hybridization with probes to detect Ptgds, Klk6, Itpr2 and Dusp1, and amplification steps. After hybridization and amplification steps to detect the mRNA of the target gene markers, the sections were blocked in 5% NDS, 0.01% Triton X100 in PBS for 30 min at RT and incubated with rat anti-MBP (AbCam, ab 7349) primary antibody 1:200 and goat anti-PDGFRα (R&D Systems, AF1062) primary antibody 1:200 in 5% NDS, 0.1% Triton X100 in PBS, O.N. at 4 °C. The following day, cells were incubated with donkey anti-rat AlexaFluor 488 conjugated (LifeTech, A21208) 1:1000 and donkey anti-goat AlexaFluor 647 conjugated (LifeTech,

A21447) 1:1000 in 5% NDS, 0.1% Triton X100 in PBS, 1 h at RT and counterstained with DAPI (1:5,000 in 0.1% Triton X100 in PBS) for 5 min. ICC washing steps were performed with 0.05% Tween-20 in PBS.

**Image acquisition**. Fluorescent images were acquired using the LSM800 confocal microscope (Zeiss). To obtain an optimal balance between the resolution of RNAscope signal and imaged area, tiled images were acquired with a ×40 water objective. The z-stack was kept to 2–3 focal planes with 4 μm step to reduce the probability of false-positive cells after image maximum projection.

**In situ sequencing**. P20 and P60 mouse coronal brain tissues and spinal cord tissues (from two mice per stage) were snap frozen in OCT mounting medium, sectioned in 10-μm-thick cryosections and stored at −80 °C until fixation. ISS, a targeted multiplexed mRNA detection assay employing padlock probes, rolling circle amplification (RCA) and barcode sequencing, was applied. Padlock probes targeting a set of genes were equipped with four-digit barcode sequences specific for each gene (5 padlock probes per gene, target sequences are supplied in Supplementary Data 2). The method and data processing were based on Ke et al.[13]. Briefly, 10-μm-thick cryosections were fixed by 3% (w/v) PFA in DEPC-treated PBS (DEPC-PBS) at room temperature (RT) for 5 min, and they were washed in DEPC-PBS followed by 0.1 N HCl treatment at RT for 5 min. The sections were washed in DEPC-PBS again, and they were subjected to reverse transcription, probe ligation and rolling circle amplification reactions using NeuroKit (1010-01, CARTANA AB, Sweden) (similar to Soldatov et al.[62]). The RCA products were detected by hybridization of AF750-labeled detection oligo (5′-UGCGU-CUAUUUAGUGGAGCC-3′, IDT Corelville, Iowa), and decoded through four cycles of barcode sequencing, each involving a base-specific incorporation of fluorescence dyes (A = Cy5, C = Texas Red, G = Cy3, T = AF488), imaging, and removing the incorporated dyes. The genes were manually selected to target the major markers for oligodendrocytes. Barcode sequences were read-out by a series of image analysis procedures.

Images were acquired using a Zeiss Axio Imager Z2 epifluorescence microscope (Zeiss Oberkochen, Germany), equipped with a ×40 objective. A series of images (10% overlap between two neighboring images) at different focal depths was obtained and the stacks of images were merged to a single image thereafter using the maximum-intensity projection (MIP) in the Zeiss ZEN software. The resulting images were then automatically stitched together into a single image containing the entire scanned area. Stitched images were used for further image analysis and aligned with the stitched images of each sequencing round. A custom-made Cellprofiler 2.2.1 pipeline was used to extract the fluorescence intensity from each of the signal (each detected RCA product) and to save all intensity information and coordinates, followed by signal decoding using a custom-made Matlab pipeline. For each signal the base with the highest fluorescence intensity was extracted and a quality score was calculated (defined as the maximum signal, divided by the sum of all signals). After thresholding, the frequency of each sequence was extracted and based on the 2D coordinates a map of genes was built as well as the signals were assigned to cells. Thereby, cell nuclei were segmented using watershed segmentation on the DAPI channel and expanded (similar to Tiklova et al.[63]). Regions of interest were drawn onto the tissue section and all signals within that region were extracted for statistical analysis. Oligodendrocyte lineage cells were identified by Sox10 and Plp1 co-labeling in the segmented cells as defined by DAPI staining, while OPCs and COPs by Sox10 co-labeling. Plp1 labeling was also observed in areas where cells were not segmented, most likely due to the presence of the transcript also in processes. For quantification (Supplementary Figs. 2 and 3), only triple or quadruple co-expression (Gene A$^+$/Gene B$^+$/Sox10$^+$/Plp1$^+$) was considered, given that the raw count numbers were very low with further degrees of co-expression, and thus not reliable, most likely due to an increase of false-negative cells for expression of individual markers (failure of detection of probes for specific genes).

**Microfiber assay and culture conditions**. Poly-L-lactic acid fibers suspended into 12-well inserts with a diameter range of 2–4 μm (Electrospinning Company, TECL006). Prior to use, inserts were soaked in 70% ethanol for 20 min, washed three times with water and coated for at least 1 h at 37 °C with 5 μg/mL PDL, followed by three washes with water to remove excess PDL and one wash with medium[11,37]. OPCs were obtained from P4 to P6 brains or spinal cords. Brains and spinal cord were removed and dissociated in single-cell suspensions using the Neural Tissue Dissociation Kit (P) (Miltenyi Biotec, 130-092-628) according to the manufacturer's protocol. OPCs were isolated by MACS with CD140a microbeads following the manufacturer's protocol (CD140a Microbead kit, Miltenyi Biotec, 130-101-547). Cells were seeded on microfibers at a density of 50,000 cells per scaffold in proliferation medium comprising DMEM/GMAX (ThermoFisher Scientific, 10565018), N2 media (ThermoFisher Scientific, 17502001), penicillin–streptomycin (ThermoFisher Scientific, 10378016), NeuroBrew (Miltenyi 130-097-263), NT-3 1 ng/ml (Peprotech, AF-450-03) and PDGF-AA 10 ng/ml (Peprotech, 315-17). After 24 h proliferation medium was changed with differentiation medium comprising DMEM/GMAX (ThermoFisher Scientific, 10565018), N2 media (ThermoFisher Scientific, 17502001), penicillin–streptomycin (ThermoFisher Scientific, 10378016), NeuroBrew

(Miltenyi 130-097-263), NT-3 1 ng/ml (Peprotech, AF-450-03), T3 40 ng/ml (Sigma T6397) and T4 40 ng/ml (Sigma 89430). Differentiation medium was changed every other day. Cells were differentiated for 3, 7 and 14 days in vitro.

**Image analysis**. Confocal images were processed with FiJi (ImageJ, NIH) to select the regions of interest (ROIs). ROI images were segmented with a customized CellProfiler pipeline. Briefly, the signals from the individual channels (DAPI, markers, GFP) were segmented; OL lineage cells (Sox10$^+$) were identified using the masking option with the DAPI counterstain. Then, the specific OL lineage populations were identified using the relate objects options with the Sox10$^+$ or Sox10$^+$-GFP$^+$ cells. Overlay images of the identified objects were exported and used to assess the percentage of cell segmentation error. Spreadsheets containing the number of parent cells (DAPI, Sox10$^+$, Sox10$^+$-GFP$^+$, Aspa$^+$, or Sox10$^+$-Aspa$^+$ cells) and child objects (Ptgds$^+$, Klk6$^+$, Egr2$^+$, Ptprz1$^+$) were exported and used to calculate the percentage of each population. Based on the average gene expression in scRNAseq dataset[5], we used a cutoff of 12, 4, 3, and 7 molecules of Ptgds$^+$, Klk6$^+$, Egr2$^+$, Ptprz1$^+$, respectively, per cell to call the analyzed OL lineage populations.

We manually assessed the percentage of error for the automated cell segmentation and attribution to 18 representative images (six images per analyzed region). We recorded a segmentation error in the identification of the nuclei of 6.40 ± 1.07% (Supplementary Fig. 1e). We did not observe any substantial error additional to the nuclei segmentation error when we identified the Sox10$^+$-GFP$^+$ nuclei by the masking option (Supplementary Fig. 1e). Our analysis is a reliable tool for the fast quantification of cells in large image datasets. We calculated the percentage of cells belonging to the oligodendrocyte lineage over the total number of cells (DAPI$^+$ nuclei) and detected the highest and lowest percentage in the corpus callosum and cortex at both P20 (65.87 ± 2.26% and 10.3 ± 0.62%) and P60 (67.95 ± 4.78% and 20.0 ± 5.13%; Supplementary Fig. 1d). This is in line with the previously described distribution of the OL lineage[64] and the relative myelination levels of the analyzed regions. The image analysis pipeline is available at the following link: https://github.com/Castelo-Branco-lab/Floriddia_et_al_2019.

**Single-cell analysis of corpus callosum**. Fast-Q alignment was performed using STAR aligner. Cells were analyzed in the single-cell analysis r-package Seurat[27,28]. Thresholds were set for a minimum of 200 genes expressed per cell, with a maximum of 3000 genes. We also set a threshold for mitochondrial expression as a percentage of total measure UMI counts, permitting cells to pass the threshold of <5% of mitochondrial UMI counts of the total number of UMI counts. We analyzed the corpus callosum of dorsal and ventral origin by individually normalizing the datasets and then performing integration in Seurat through canonical correlation analysis. We clustered cells using the Leiden clustering algorithm with the resolution parameter set at 0.8, using 30 principal components on 3000 variable genes, resulting in 11 clusters. We then proceeded to find markers for individual clusters using the FindAllMarkers function, with a cutoff of 0.25-fold change required.

**Single-cell analysis of spinal cord tissue**. Fast-Q alignment was performed using STAR aligner. Cells were analyzed in the single-cell analysis r-package Seurat[27,28]. Thresholds were set for a minimum of 500 genes expressed per cell, with a maximum of 7000 genes. We also set a threshold for mitochondrial expression as a percentage of total measure UMI counts, permitting cells to pass the threshold of <10% of mitochondrial UMI counts of the total number of UMI counts. We analyzed the injury site, Wallerian degeneration, and control samples by combined normalization of the datasets without integration. This was necessary because of dataset-specific cell populations making integration more difficult. We clustered cells using the Leiden clustering algorithm with the resolution parameter set at 0.6, using 30 principal components on 3000 variable genes, resulting in 13 clusters. We then proceeded to find markers for individual clusters using the FindAllMarkers function, with a cutoff of 0.25-fold change required.

**Subclustering of spinal cord cells**. We performed subclustering of cells in the spinal cord datasets, to infer the possible identities of small clusters. We subclustered OPC and unknown clusters using a previously published approach GeneFocus[6] utilizing the spatial statistic Morans I on a cell manifold layout. We adjusted this approach to be more robust and precise by inferring spatial proximity using a three-dimensional UMAP[65], and approximating spatial significance through the use of Monte-Carlo simulation (SPDEP package, https://doi.org/10.1007/s11749-018-0599-x). After an iterative sequence of rounds of spatial filtering, we used the feature set to guide a hierarchical clustering to subcluster the cells into a variety of populations, including Schwann cells, ependymal cells, immune cells, neurons and oligodendrocytes. We used a combination of results from the marker selection function in Seurat, together with our 26 iterations filtered genelist (~200 genes) to provide specific and robust markers that aided the clustering. Code can be found at https://github.com/Castelo-Branco-lab/Floriddia_et_al_2019.

**Label transfer**. We transferred the labels of a previously published dataset on oligodendrocyte heterogeneity[5], to attempt to relate the found clusters to previous findings of canonical oligodendrocyte clusters. We used the label TransferData

function in Seurat on both the corpus callosum and spinal cord datasets individually to transfer the labels, using the first 15 principal components on the correlated feature-space.

**Calculation of the frequency ratios of clusters**. We converted the frequencies of cells for each cluster to a percentage by column normalization of the total cells from each sample, making the populations comparable through integration into a common plot, such as the stacked bar plot and dotplots. The change between populations was measured in absolute percentage and plotted in a heatmap between the populations.

**2D-Vulcano plots**. To illustrate multidimensional comparisons, we performed differential expression using Wilcoxon-rank sum-test within the mature oligodendrocyte populations. Results from both the comparison between injury site and Wallerian degeneration were plotted with an additional comparison between the control condition, and the non-control conditions. Log Fold changes were plotted on each respective axis of a scatterplot, generating a 2D-Vulcano plot. For each gene, we chose the lowest $p$ value for both comparisons to include in the plot.

**Reactome pathways**. Reactome pathways were calculated for each comparison using the r-package ReactomePA[66]. We included pathway analysis for both up- and downregulated genes, as well as pathways enriched in all conditions per OL cluster. We embedded the pathway results in a 2D-Vulcano plot, by averaging the log fold change for all genes in the pathway, from the log fold change of the comparisons that generated the 2D-Vulcano plot of the individual genes.

**Statistics**. Statistics on the spatial distribution of the OL lineage populations was performed using two-way ANOVA. For multiple comparison analysis, the Sidak's correction was applied.

Differential gene expression analysis was performed using pairwise Wilcoxon-rank sum-tests using the stats package in R, on averaged expression per cluster. Significant genes were selected with an FDR adjusted $p$ value < 0.01. The heatmap displays the most top 20 highest enriched genes as measured by z score.

**Reporting summary**. Further information on research design is available in the Nature Research Reporting Summary linked to this article.

## Data availability
We declare that the data supporting the findings of this study are available within the paper. Single-cell RNA Sequencing dataset is available in GEO (accession number GSE128525). The sequencing dataset can also be explored at and visualized at https://ki.se/en/mbb/oligointernode. Additional information is available upon reasonable request to the authors. Source data are provided with this paper.

## Code availability
The code for cell clustering analysis and the image analysis pipeline is available at the following link: https://github.com/Castelo-Branco-lab/Floriddia_et_al_2019.

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

## Acknowledgements

We would like to thank Sarah Foerster, Richa B. Tripathi, William D. Richardson, and Robin J.M. Franklin for the fruitful discussions and sharing the Emx1::Cre-Sox10::Cre-LoxP-GFP-STOP-TdTom line, Antonio del Sol and Srikanth Ravichandran for discussions, Rashid Holtinkoski, Eneritz Agirre, Ana Mendanha Falcão, Alessandra Nanni, Johnny Söderlund, Ahmad Moshref, Tony Jimenez-Baristain, Lesley Kirby, the Facility Management and Administration at Biomedicum (Karolinska Institutet, Stockholm) for laboratory management and support. Dr. Göran Månsson and Connla Edwards at the Biomedicum Imaging Core Facility (Karolinska Institutet, Stockholm), Dr. Jaromir Mikes at the Mass Cytometry National Facility (SciLife Lab, Stockholm); Drs. Karolina Wallenborg, Anna Juréus, Marcela Ferella at the Eukariotic Single Cell Genomics facility (SciLife Lab, Stockholm) "In situ sequencing" unit of the spatial-omics infrastructure (SciLifeLab, Stockholm), for technical assistance with the ISS experiments; Katarina Ericsson, Kristoffer Tenebro Berglund, Johanna Hornstrand at the Comparative Medicine Biomedicum facility (Karolinska Institutet, Stockholm) and their respective facility managements, the National Genomics Infrastructure and Uppmax for providing assistance in massive parallel sequencing and computational infrastructure. The bioinformatics computations were performed on resources provided by the Swedish National Infrastructure for Computing at UPPMAX, Uppsala University. NIH Pathway to Independent Award (5K99EY029330) supported L.O.S. T.L. was supported by a Ph.D. grant (SFRH/BD/114731/2016) and M.G. was supported by grant POCI-01-0145-FEDER-029516, cofinanced by the ERDF/FEDER under the framework Competitiveness and Internationalization Operational Program (POCI) and funds from FCT/'Ministério da Ciência, Tecnologia e Ensino Superior' (FCT/MCTES) through the Portuguese State Budget. Work in G.C.-B.'s research group was supported by Swedish Research Council (grant 2015-03558 and 2019-01360), European Union (Horizon 2020 Research and Innovation Programme/ European Research Council Consolidator Grant EPIScOPE, grant agreement number 681893), Swedish Brain Foundation (FO2017-0075), Knut and Alice Wallenberg Foundation (grant 2019-0107), The Swedish Society for Medical Research (SSMF, grant JUB2019), Ming Wai Lau Centre for Reparative Medicine, Strategic Research Programme in Neuroscience (StratNeuro) and Karolinska Institutet.

## Author contributions

E.M.F. and G.C.-B. conceived the project, designed the study and interpreted results. E.M.F., T.L., S.Z., M.M.H., M.A., P.K. performed experiments and analyzed data. J.P.G.S., C.Y., S.B.M. and E.L.B. performed experiments. D.v.B. analyzed data. E.M.F., T.L., S.Z., D.v.B., P.K., M.M.H., E.L.B., M.G., L.O.S., J.F., M.N., and G.C.-B. discussed the results of the study. E.M.F. and G.C.-B. wrote the manuscript with feedback from all co-authors.

## Funding

## Competing interests

M.N. and M.M.H. held shares in Cartana AB, a company commercializing in situ sequencing reagents. E.M.F. is currently employed as an Associate Editor of *Nature Communications*. She was not involved in the decision-making process for this manuscript and did not have access to confidential information pertaining to peer review and editorial process. The remaining authors declare no competing interests.
