## [Peer Review File · Nature Communications]

Reviewers' comments:

Reviewer #1 (Remarks to the Author):

The authors have seriously considered and addressed the previous comments in their revised manuscript. They have added many new datasets including ISS, TFEB cko mice, in vitro myelination experiments, and sc-RNA-seq of the injured spinal cord to clarify some of the uncertainties in the previous version. The description of the similarities and differences between the MOL2 and MOL5/6 types of cells is much more clearly laid out. The spinal cord injury experiments are still quite complicated, but the main points come across more clearly now.

The most interesting addition to me is the microfiber myelination experiments, as this addresses functional differences between the MOL2 and MOL5/6 subtypes. The differential ability of MOL2 and MOL5/6 cells to survive in the spinal cord cultures and myelinate microfibers is very interesting. This property may also be related to the lower contribution of MOL2 after spinal cord injury. Have the authors tested different diameter fibers? Since axons in the gray matter are smaller than those in the ascending dorsal column, perhaps MOL2 cells in the white matter only myelinate larger diameter axons and die when they only see smaller diameter axons. Since the Ptgds+ MOL5/6 type of cells only account for <50% of the MBP+ cells, I am wondering what the remaining MBP+ cells are. Have the authors tried using other markers of the MOL2 type of cells to make sure that the culture condition had not caused downregulation of Klk6? Alternatively, does culture induce genes of the MOL1 subtype?

For the TFEB conditional knockout studies in Fig. 4, it would be better to show the abundance of total Aspa+ cells, MOL2 Aspa+ cells, and MOL5/6 Aspa+ cells, along with the percentage. This would reveal any effects that might be masked by the unequal denominator.

Overall, the manuscript is much improved and contains interesting new information that will provide an important foundation for future analysis of the heterogeneity among oligodendrocytes. The presentation of the data has been greatly improved, so that the important points can be easily followed now with the figures in the main manuscript.

Reviewer #2 (Remarks to the Author):

The authors have addressed my initial comments when this manuscript was submitted to Nature Neuroscience. I support publication after reformatting Figure 6. The color scale in panel C would be easier to read if it were divergent and centered on zero (e.g. blue to white to red) to make it clear which changes are negative and positive. Panel E would be easier to interpret if you colored gene symbols to indicate which cell type they are markers of. The P-values of labeled genes are all highly significant so those colors do not add much to the plot. Panel F is difficult to read with small fonts and many callout lines. Another option is to use a Cleveland dot plot where GO categories are listed on the left as rows, and fold changes are shown as dots on the right.

Reviewer #3 (Remarks to the Author):

The authors have addressed most points raised by the reviewer, but there are still issues that need to be improved. Most of these issues are related to the presentation of the data. The authors added in situ sequencing data, but these data poorly described and illustrated, it not easy to follow their conclusions.

General comments:

Many of the figures are unreadable, for example the labeling of the graphs in Figure 3F,G,D; Figure 5, Figure S2, 3, 5

Some of the images are of low quality, for example Figure S3C

The discussion can be improved. Many interesting aspects of the paper are not discussed and some statements are unclear, for example, what are "OPC intrinsic mechanism" or what is "a considerable presence"

Specific comments:

Figure 1: The authors never show the overlap between Klk6 as a MOL2 marker and Ptgd as a MOL5/6 marker. How much overlap is observed between these two populations? This information is important to evaluate the segregation of these two subtypes. Information on the overlap is only provided for the disease model.

Figure 1H: no Sox10 labeling is seen

Figure 3C: it looks like GFP and tdTomato derived cells segregate. Could the authors comment?

Figure 4: The authors present the percentage of MOL2 and 5/6 overlapping with Aspa. They should also present the total amount of Aspa cells in Tfeb knockout vs wild-type.

Figure 5F and G: No Klk6 signal is seen in the images. This does not match the quantification. The symbols in the graph are barely readable.

Supplemental Figure 2: The authors state in the text page 4: MOL2 express Anxa, Hopx, and Klk6 and refer to Figure S2, but it is difficult to find the data in the Figure. In F these markers are only partially shown, and co-localization cannot be seen. The same is true for MOL5/6. The data shown in C to E is not understandable. The symbols for brain and spinal cord are too small, and it is not clear how the authors come to the conclusion that Anxa and Hopx are new markers for MOL2, while Grm3 and Car2 are markers for MOL5/6

We would like to thank the reviewers for their insightful comments. We have now introduced further changes in the manuscript addressing these comments (highlighted in yellow). We hope that the reviewers find our revised version of our manuscript suitable for publication in Nature Communications.

Reviewer #1 (Remarks to the Author):

The authors have seriously considered and addressed the previous comments in their revised manuscript. They have added many new datasets including ISS, TFEB cko mice, in vitro myelination experiments, and sc-RNA-seq of the injured spinal cord to clarify some of the uncertainties in the previous version. The description of the similarities and differences between the MOL2 and MOL5/6 types of cells is much more clearly laid out. The spinal cord injury experiments are still quite complicated, but the main points come across more clearly now.

We would like to thank the reviewer for considering that we have addressed the previous comments and that the new datasets clarify several aspects of the manuscript.

The most interesting addition to me is the microfiber myelination experiments, as this addresses functional differences between the MOL2 and MOL5/6 subtypes. The differential ability of MOL2 and MOL5/6 cells to survive in the spinal cord cultures and myelinate microfibers is very interesting. This property may also be related to the lower contribution of MOL2 after spinal cord injury. Have the authors tested different diameter fibers?

We agree with the reviewer and we also hypothesized that the fiber diameter might play an important role in guiding the maturation of OLs in culture. Indeed, the dorsal funiculi of the spinal cord have large diameter sensory axons and MOL2 are a prominent OL population. In the current manuscript, we used fibers with a scaffold containing a mix of fibers with thickness of 2-4 μm , with the idea to analyze whether MOL2 and MOL5/6 preferentially myelinate large and small diameter fibers, respectively. Nevertheless, as reported in the manuscript, we have not observed Klk6+/MBP+ oligodendrocytes in our 3D cultures. We have now clarified the range of fiber diameters in the main text and in the methods section.

Since axons in the gray matter are smaller than those in the ascending dorsal column, perhaps MOL2 cells in the white matter only myelinate larger diameter axons and die when they only see smaller diameter axons. Since the Ptgds+ MOL5/6 type of cells only account for <50% of the MBP+ cells, I am wondering what the remaining MBP+ cells are. Have the authors tried using other markers of the MOL2 type of cells to make sure that the culture condition had not caused downregulation of Klk6? Alternatively, does culture induce genes of the MOL1 subtype?

In order to further characterize the composition of oligodendrocyte lineage cells in the microfiber cultures, we have now performed ICC for PDGFRA and RNAscope for *Itpr2* (newly formed oligodendrocyte marker) and *Dusp1*, a marker of MOL1, after 3 and 14 days. We have included

this new data in the manuscript. We observed that PDGFRA⁺ OPCs were approximately 15% of the cells at 3 days and decrease considerably at 14 days (new Supplementary Fig. 6D,G). We also observed that 40% of the cells in brain derived cultures at 3 days were NFOLs (*Itrp2*⁺), with an increase to nearly 60% at 14 days (new Supplementary Fig. 6F, G). 40-60% of the MBP⁺ cells were *Itrp2*⁺ (new Supplementary Fig. 6F,H), indicating that a large proportion of OLs had not transitioned to a MOL state. When we quantified the percentage of *Dusp1*⁺- MBP⁺(MOL1) and *Klk6*⁺- MBP⁺ (MOL2) cells wrapping the microfibers, we did not observe any *Klk6*⁺- MBP⁺ cells in culture (new Supplementary Fig. 6G, H). While we could detect rare *Dusp1*⁺-MBP⁺ cells at 14 days (new Supplementary Fig. 6E, H), the vast majority of *Dusp1*⁺ were not MOLs. Thus, around 40% *Ptgds*⁺- MBP⁺ cells in both brain and spinal cord OL lineage cultures (Fig.4F-I). These data suggest that OPCs mainly differentiate into NFOLs and MOL5/6, but not to MOL1 or MOL2 in culture conditions used.

For the TFEB conditional knockout studies in Fig. 4, it would be better to show the abundance of total Aspa⁺ cells, MOL2 Aspa⁺ cells, and MOL5/6 Aspa⁺ cells, along with the percentage. This would reveal any effects that might be masked by the unequal denominator.

To enhance data transparency and further clarify the distribution and density of the Aspa⁺, MOL2 Aspa⁺, and MOL5/6 Aspa⁺ cells we are now showing a table reporting both cell numbers and percentages as summary table (new Fig. 4E), in a similar fashion as for the quantification reported in Fig. 1M.

Overall, the manuscript is much improved and contains interesting new information that will provide an important foundation for future analysis of the heterogeneity among oligodendrocytes. The presentation of the data has been greatly improved, so that the important points can be easily followed now with the figures in the main manuscript.

We would like to thank the reviewer for considering our manuscript much improved and an important foundation for future studies. We hope that the reviewer finds our revised version of our manuscript suitable for publication in Nature Communications.

Reviewer #2 (Remarks to the Author):

The authors have addressed my initial comments when this manuscript was submitted to Nature Neuroscience. I support publication after reformatting Figure 6. The color scale in panel C would be easier to read if it were divergent and centered on zero (e.g. blue to white to red) to make it clear which changes are negative and positive. Panel E would be easier to interpret if you colored gene symbols to indicate which cell type they are markers of. The P-values of labeled genes are all highly significant so those colors do not add much to the plot. Panel F is difficult to read with small fonts and many callout lines. Another option is to use a Cleveland dot plot where GO categories are listed on the left as rows, and fold changes are shown as dots on the right.

We have now introduced the changes suggested by the reviewer in Panel C and Panel E. We kept Panel 6F, so the reader can compare to Panel 6 E, but changed the size of the text for readability. We also include now a Cleveland dot plot in Figure S8.

Reviewer #3 (Remarks to the Author):

The authors have addressed most points raised by the reviewer, but there are still issues that need to be improved. Most of these issues are related to the presentation of the data. The authors added in situ sequencing data, but these data poorly described and illustrated, it not easy to follow their conclusions.

General comments:

Many of the figure are unreadable, for example the labeling of the graphs in Figure 3F,G, D; Figure 5, Figure S2, 3, 5 – increase graph size in all

As requested by the reviewer, we have increased the size of the font and the overall size of the graphs in Figure 3F-G, J, D, Figure 5D-E, H-I, J-K, P-Q, Figure S2, Figure S3 and Figure S5.

Some of the images are of low quality, for example Figure S3C

The images are of high quality, but we agree that their small format limits their readability, so we have increased their size and brightness.

The discussion can be improved. Many interesting aspects of the paper are not discussed and some statements are unclear, for example, what are “OPC intrinsic mechanism” or what is “a considerable presence”

We thank the Reviewer for the suggestion. We have edited the discussion in few critical paragraphs, which now read:

Additionally, we also show that the generation of distinct MOL populations is independent of the OPC intrinsic mechanisms, such as their time and domain of origin during development or programmed cell death^{28,30,35}.

Here, we also report that the MOL2 population is particularly affected by traumatic injury, as this population is lost at the injury site and it does not repopulate this region even five months after injury, a time point when the physiological partial remyelination and repair has occurred. Furthermore, we observed that MOL2 and MOL5/6 are still quite abundant in regions where the axons forming the dorsal funiculi are lost, as they underwent Wallerian degeneration (SFig. 7 D-E).

“Here we show that axonal degeneration does not affect the relative contribution of MOL2 and MOL5/6 to the OL lineage, suggesting that MOLs have stable identities although their surrounding axons are degenerating or lost.”

Specific comments:

Figure 1: The authors never show the overlap between Klk6 as a MOL2 marker and Ptgds as a MOL5/6 marker. How much overlap is observed between these two populations? This information is important to evaluate the segregation of these two subtypes. Information on the overlap is only provided for the disease model.

Klk6 (MOL2) and Ptgds (MOL5/6) were identified as markers in our previous publication (Marques et al., Science 2016). We now include in Fig. S1J the single-cell RNA-seq expression profiles of genes enriched in MOL2, MOL5 and MOL6 in the oligodendrocyte lineage in the juvenile and adult CNS at a single cell level, as in Marques et al, 2016. We also include now density plots for the expression of Klk6 and Ptgds in the MOL2, MOL5 and MOL6 populations, in the spinal cord single cell RNA-seq data (Supplementary Fig. 8 G). These density plots depict the probability of finding a cell at a determined expression level (at the x-axis) and show that there is a high density of MOL2 with low expression of Ptgds (undetectable by RNASCOPE and ISS) and upon increasing counts of Ptgds, the probability of find MOL2 sharply reduces, while the probability to find MOL5 or MOL6 increases, indicating that Klk6 and Ptgds expression indeed allows the segregation of the 2 subtypes.

We have reported the information on the overlapping population for the disease model due to the limited number of fluorophores we could use for each slide and the experimental design we chose. Indeed, for the spinal cord injury models, we did not fate map OPCs, therefore we had the opportunity to stain for Sox10, Klk6, and Ptgds markers on the same section, while for the P20 and P60 time points, we used tissue from Pdgfr-alpha::CreERT1 – GFP fl/fl mice we have injected with tamoxifen to induce recombination and fate map OPCs. In this tissue, we could stain for GFP, Sox10 and Klk6 or Ptgds markers. However, our staining for the disease model and scRNAseq of juvenile and adult OL lineage showed that the number of OLs expressing both Klk6 and Ptgds is limited, suggesting that MOL2 and MOL5/6 segregate well.

Figure 1H: no Sox10 labeling is seen

We believe that the Reviewer might have confused the Sox10 and Klk6 labeling in Fig. 1H. Indeed, in Figure 1H, Sox10 and Klk6 were stained using fluorophores Cy3 (red channel) and Cy5 (far red channel – in white), respectively, while in Figure 1 B, E, K Sox10 is stained using Cy5 (fare red channel – in white). Therefore, in Figure 1H. Sox10 is clearly visible, while Klk6 is absent. Indeed, in the corpus callosum, the MOL2 population is almost absent, as quantified in Fig. 1I and M.

Figure 3C: it looks like GFP and tdTomato derived cells segregate. Could the authors comment?

The GFP and tdTomato cells are intermingled. We have now reduced the size of the dots so the reader can analyse the figure in a better way.

Figure 4: The authors present the percentage of MOL2 and 5/6 overlapping with Aspa. They should also present the total amount of Aspa cells in Tfeb knockout vs wild-type.

We are now showing a table reporting both cell numbers of Aspa+ cells as well as MOL2 and MOL5/6 and percentages as summary table (new Fig. 4E) we quantified in the TFEB cKO and littermate controls, in a similar fashion as for the quantification reported in Fig. 1M.

Figure 5F and G: No Klk6 signal is seen in the images. This does not match the quantification. The symbols in the graph are barely readable.

We have increased the labeling of the graphs in Figure 5D-E, H-I, J-K, P-Q. Regarding the representative images, we would like to point out that the quantification in panel K shows variability in the % of MOL2 cells at the injury site.

Supplemental Figure 2: The authors state in the text page 4: MOL2 express Anxa, Hopx, and Klk6 and refer to Figure S2, but it is difficult to find the data in the Figure. In F these markers are only partially shown, and co-localization cannot be seen. The same is true for MOL5/6. The data shown in C to E is not understandable. The symbols for brain and spinal cord are too small, and it is not clear how the authors come to the conclusion that Anxa and Hopx are new markers for MOL2, while Grm3 and Car2 are markers for MOL5/6

Anxa and Hopx (MOL2), Grm3 and Car2 (MOL5/6) were not identified as markers in the current manuscript, but rather in our previous publication (Marques et al., Science 2016). We apologize for not being clear in the manuscript, we have now included in Fig. S1J the single-cell RNA-seq expression profiles of genes enriched in MOL2, MOL5 and MOL6 in the oligodendrocyte lineage in the juvenile and adult CNS at a single cell level, as in Marques et al, 2016. We also clarify this in the main text in the part of the ISS.

The images in Figure S2F (now S2C) are taken at low magnification and thus were not intended for co-localization purposes, for which the images Figure S2D (previous Figure S2G) are better suited. The images in Figure S2F (now S2C) were used for quantification and to illustrate the overall detection levels of some of the investigated markers. The number of cells double positive for *Anxa5/Klk6* or *Anxa5/Hopx* (MOL2) in the brain is lower than in the spinal cord, unlike cells positive for *Ptgds/Car2* and *Ptgds/Grm3* (MOL5/6). We have changed the main text to clarify this point:

We took advantage of the higher multiplexing power of ISS compared to RNAscope ISH to detect OPC/COP, MOL1, MOL2, and MOL5/6 based on the combination of multiple marker genes (Supplementary Fig. 2, 3 and Supplementary Table 1)-. MOL2 are enriched in Anxa5, Hopx, and Klk6 and MOL5/6 have increased expression of Pdgfra, and also express Grm3 and Car2 (Supplementary Fig. 1J)³. ISS indicates the number of Sox10⁺/Plp1⁺ cells also positive for Anxa5/Klk6 or Anxa5/Hopx (MOL2) in the brain is lower than in the spinal cord, unlike Sox10⁺/Plp1⁺ cells positive for Pdgfra/Car2 and Pdgfra/Grm3 (MOL5/6) (Supplementary Fig. 3A, Supplementary Table 1-2). Furthermore, we observed that MOL5/6 are also increased with age in

the cortex and corpus callosum, and spinal cord (Supplementary Fig. 3A, B, Supplementary Table 1-2).

We have also explained in more detail Supplementary Fig. 3C in the main text:

Also in this instance, we confirmed the MOL spatial preference by ISS analysis. Indeed, *Ptgds*⁺/*Sox10*⁺/*Plp1*⁺ or *Car2*⁺/*Sox10*⁺/*Plp1*⁺ cells (MOL5/6) are present in higher proportions in GM compared to WM, while *Anxa*⁺/*Sox10*⁺/*Plp1*⁺ or *Klk6*⁺/*Sox10*⁺/*Plp1*⁺ cells (MOL2) are more abundant in the WM compared to the GM of the spinal cord, particularly at P20 (Supplementary Fig. 3C, Supplementary Table 1).

We have now split and increase the size of the previous Figure S2C-E (now Figure S2E and Figures S3A-C) so the symbols for brain and spinal cord can be read.

REVIEWERS' COMMENTS

Reviewer #1 (Remarks to the Author):

The manuscript is much improved with the new data and revised text.

The new quantification for the OPC-microfiber coculture clarifies the findings, and the authors' interpretation makes sense.

The authors have added a table in Figure 4E to show the density of oligodendrocytes in Tfeb conditional knockout. The data reveals a greater complexity of the Tfeb story than was suggested in the prior reports by the Sun and Talbot groups and suggests that TFE8 may not be uniformly pro-apoptotic, as the number of ASPA+ cells in gray matter is reduced. More importantly, when comparing to the degree of reduction or increase of MOL5/6 or MOL2 cells versus that of ASPA+ cells in gray and white matter, the magnitude of change in the MOL subtypes is greater than that of total ASPA+ cells (e.g. 2-fold reduction of ASPA+ cells in gray matter but 4-fold reduction of MOL5/6). While the authors need not make a big deal of the differential effects of Tfeb cko in gray and white matter, they should be more accurate in stating what is relevant here; that there is a more exaggerated representation of MOL2 and MOL5/6. in the mutant. This should be stated in the paragraph and not just dismissed as no change, since the new density data suggests that the MOLs (as opposed to other cell types) are what is changing prominently in the knockout. The sentences on lines 231 and 233 "We did not observe changes in the contribution of MOL2... or MOL5/6..." should be rephrased to reflect this. The more precise description of the more exaggerated contribution of the different MOL types will be consistent with the general conclusion. Adding the more accurate description of the results will not deter from the main point, but it will reinforce it.

Very minor points.

The following sentence in the Introduction is a bit difficult to understand and not quite accurate. cortical electrical impulses are synchronized because the conduction time between the left and right brain hemispheres is 30 ms and 150-300 ms through unmyelinated short projecting fibers. 1) maybe the "and" between the two phrases should be "whereas it would take 150-300 ms to conduct through unmyelinated....".

2) This is an interesting statement but does not necessarily mean that electrical impulses are "synchronized". It means that the speed of conduction is greatly increased with myelin. Maybe there is something I am missing? This can be rephrased to avoid confusion.

Line 130 – tracks should be tracts (fiber tracts)

I suggest the authors go over the manuscript carefully to correct any vague or inaccurate phrases.

The manuscript is now significantly improved, and the main points come across more clearly.

Reviewer #3 (Remarks to the Author):

The authors have answered all points raised. I recommend publication.

REVIEWERS' COMMENTS

Reviewer #1 (Remarks to the Author):

The manuscript is much improved with the new data and revised text.

The new quantification for the OPC-microfiber coculture clarifies the findings, and the authors' interpretation makes sense.

We thank Reviewer #1 for their positive feedback.

The authors have added a table in Figure 4E to show the density of oligodendrocytes in Tfeb conditional knockout. The data reveals a greater complexity of the Tfeb story than was suggested in the prior reports by the Sun and Talbot groups and suggests that TFEB may not be uniformly pro-apoptotic, as the number of ASPA+ cells in gray matter is reduced. More importantly, when comparing to the degree of reduction or increase of MOL5/6 or MOL2 cells versus that of ASPA+ cells in gray and white matter, the magnitude of change in the MOL subtypes is greater than that of total ASPA+ cells (e.g. 2-fold reduction of ASPA+ cells in gray matter but 4-fold reduction of MOL5/6). While the authors need not make a big deal of the differential effects of Tfeb cko in gray and white matter, they should be more accurate in stating what is relevant here; that there is a more exaggerated representation of MOL2 and MOL5/6. in the mutant. This should be stated in the paragraph and not just dismissed as no change, since the new density data suggests that the MOLs (as opposed to other cell types) are what is changing prominently in the knockout. The sentences on lines 231 and 233 "We did not observe changes in the contribution of MOL2... or MOL5/6..." should be rephrased to reflect this. The more precise description of the more exaggerated contribution of the different MOL types will be consistent with the general conclusion. Adding the more accurate description of the results will not deter from the main point, but it will reinforce it.

We thank Reviewer #1 for carefully revising the new data added in the manuscript. As they mention, the different effect of the cKO on Aspa+ cells and MOL5/6 is quite dramatic in the GM of the spinal cord. However, we find that the difference in the WM and for the MOL2 populations are more subtle. We want to acknowledge this important point without overstating our findings, therefore we have revised the text as follows:

When we analyzed the effect of the deletion of TFEB on the MOL2 and MOL5/6 populations, the number of Aspa+ cells, MOL2 and MOL5/6 suggest a different representation of MOLs in the mutant mice (Fig. 4E). This was particularly evident in the grey matter of the Olig2::Cre+TFEB^{fl/fl} spinal cord tissue. Indeed, here we observed a low percentage of MOL5/6 and more prominent reduction compared to Aspa+ cells (Fig. 4A-B, D, E). To note, the GM of the spinal cord is a region where MOL5/6 are very abundant in WT (Fig. 2G, H) as well as in Olig2::Cre-TFEB^{fl/fl} control littermate mice (Fig. 3A-B, D). The percentage of the MOL5/6 calculated over the Aspa+ cells in the WM of Olig2::Cre+TFEB^{fl/fl} and Olig2::Cre-TFEB^{fl/fl} spinal cord did not show statistically significant changes (Fig. 4D, E). Similarly, the percentage of the MOL2 over the Aspa+ cells in GM and WM of the spinal cord was comparable between genotypes (Fig. 4A-C, E). Our results suggest that TFEB-dependent programmed cell death of premyelinating OLs does not have a regulatory role in the selection of specific MOL populations in the spinal cord.

Very minor points.

The following sentence in the Introduction is a bit difficult to understand and not quite accurate. cortical electrical impulses are synchronized because the conduction time between the left and right brain hemispheres is 30 ms and 150-300 ms through unmyelinated short projecting fibers.

1) maybe the “and” between the two phrases should be “whereas it would take 150-300 ms to conduct through unmyelinated....”.

2) This is an interesting statement but does not necessarily mean that electrical impulses are “synchronized”. It means that the speed of conduction is greatly increased with myelin.

Maybe there is something I am missing? This can be rephrased to avoid confusion.

We edited the text as follows:

For instance, cortical electrical impulses can be synchronized. Indeed, the conduction time between the left and right brain hemispheres is 30 ms, while it is 150-300 ms through unmyelinated short projecting fibers in the same hemisphere, so that impulses from different neurons may reach their target neurons in a coordinated manner when needed.

Line 130 – tracks should be tracts (fiber tracts)

The typo has been corrected.

I suggest the authors go over the manuscript carefully to correct any vague or inaccurate phrases.

The manuscript is now significantly improved, and the main points come across more clearly.

We thank Reviewer #1 for their suggestion and positive feedback. We have revised the text to improve clarity.

Reviewer #3 (Remarks to the Author):

The authors have answered all points raised. I recommend publication.

We thank Reviewer #3 for their positive feedback.